# Loss of GFAT-1 feedback regulation activates the hexosamine pathway that modulates protein homeostasis

Sabine Ruegenberg [1,2,5], Moritz Horn [1,5], Christian Pichlo [2], Kira Allmeroth [1], Ulrich Baumann [2]* & Martin S. Denzel [1,3,4]*

Glutamine fructose-6-phosphate amidotransferase (GFAT) is the key enzyme in the hexosamine pathway (HP) that produces uridine 5′-diphospho-N-acetyl-D-glucosamine (UDP-GlcNAc), linking energy metabolism with posttranslational protein glycosylation. In *Caenorhabditis elegans*, we previously identified *gfat-1* gain-of-function mutations that elevate UDP-GlcNAc levels, improve protein homeostasis, and extend lifespan. GFAT is highly conserved, but the gain-of-function mechanism and its relevance in mammalian cells remained unclear. Here, we present the full-length crystal structure of human GFAT-1 in complex with various ligands and with important mutations. UDP-GlcNAc directly interacts with GFAT-1, inhibiting catalytic activity. The longevity-associated G451E variant shows drastically reduced sensitivity to UDP-GlcNAc inhibition in enzyme activity assays. Our structural and functional data point to a critical role of the interdomain linker in UDP-GlcNAc inhibition. In mammalian cells, the G451E variant potently activates the HP. Therefore, GFAT-1 gain-of-function through loss of feedback inhibition constitutes a potential target for the treatment of age-related proteinopathies.

[1] Max Planck Institute for Biology of Ageing, 50931 Cologne, Germany. [2] University of Cologne, Institute of Biochemistry, 50674 Cologne, Germany. [3] CECAD—Cluster of Excellence, University of Cologne, 50931 Cologne, Germany. [4] Center for Molecular Medicine Cologne (CMMC), University of Cologne, 50931 Cologne, Germany. [5] These authors contributed equally: Sabine Ruegenberg, Moritz Horn. *email: ulrich.baumann@uni-koeln.de; martin.denzel@age.mpg.de

A progressive decline of physiological functions limits healthspan and survival in most organisms. The aging process is modulated by specific signaling pathways and their manipulation can result in lifespan extension, suggesting that aging is a coordinated process. Relevant pathways include insulin/IGF-1 signaling, mTOR, and the AMPK cascade[1]. While the plasticity of the aging process suggests that pathways could be targeted to prolong life and postpone age-related diseases in humans, few drug candidates have emerged. For example rapamycin, which targets mTOR, and metformin that acts through yet incompletely understood mechanisms, have entered clinical trials with the goal to slow aging and prevent age-related phenotypes[2,3]. Thus, there is a clear need for the identification of druggable targets and respective drugs in gerontology.

The ubiquitous hexosamine pathway (HP) is essential for aminosugar biosynthesis (Fig. 1a). Increased HP activity extends lifespan and ameliorates pathology in multiple proteotoxic disease models in the nematode *Caenorhabditis elegans*[4]. The HP converts the glycolysis intermediate fructose-6-phosphate (Frc6P) to uridine 5′-diphospho-*N*-acetyl-D-glucosamine (UDP-GlcNAc), using between 1% and 3% of total cellular glucose[5]. UDP-GlcNAc is a precursor for several important biomolecules like glycosaminoglycans and is an essential substrate for protein glycosylation reactions in mammals. Mucin-type O-glycosylation plays an important role in the extracellular matrix and N-glycosylation contributes to cellular protein homeostasis by governing the protein folding process in the endoplasmic reticulum (ER)[6,7]. Additionally, GlcNAc can be transferred as a single moiety to serine or threonine residues of target proteins[8]. This so-called O-GlcNAcylation competes with phosphorylation and fine tunes multiple processes governing cellular physiology[9,10]. Interestingly, O-GlcNAcylation potentially plays a critical role in several neurodegenerative disorders[11]. In Alzheimer's disease, hyperphosphorylation of tau protein triggers aggregation, which is ameliorated by increasing O-GlcNAcylation[12,13]. Furthermore, HP activation induces a proteoprotective cellular program in *C. elegans* and in mice through mechanisms that are not yet fully understood[4,14]. Interestingly, specific single amino acid substitutions in *C. elegans* glutamine fructose-6-phosphate amidotransferase-1 (GFAT-1, EC 2.6.1.16), which is the rate-limiting enzyme of the HP, result in gain-of-function and in significantly increased cellular UDP-GlcNAc levels that lead to significant lifespan extension[4].

In the first step of the HP, GFAT synthesizes D-glucosamine-6-phosphate (GlcN6P) from L-glutamine (L-Gln) and Frc6P, releasing L-glutamate (L-Glu)[15]. GFAT contains two domains: the glutaminase domain responsible for releasing ammonia in the hydrolysis of L-Gln to L-Glu and the isomerase/transferase domain, which catalyzes both the isomerization of Frc6P to D-glucose-6-phosphate (Glc6P) as well as the transfer of ammonia to Glc6P to produce GlcN6P[16]. The active sites of the two domains are linked by an ammonia channel, which forms when GFAT engages with both of its substrates, L-Gln and Frc6P[17]. GFAT amino acid sequence and biochemical function are conserved from bacteria to humans, but only eukaryotic GFAT is inhibited by UDP-GlcNAc[18–20]. Moreover, eukaryotic GFAT is modulated through phosphorylation by cAMP-dependent protein kinase (PKA) and AMP-activated protein kinase (AMPK). However, the effects of phosphorylation on GFAT activity are controversially discussed in the field[21–25]. Two eukaryotic GFAT paralogs exist, GFAT-1 and GFAT-2, which show 75–80% amino acid sequence identity in mice and humans, and primarily differ in their tissue-specific expression patterns[26].

Most insights into the GFAT structure and function come from its bacterial homolog, glucosamine-6-phosphate synthase (GlmS), whose full-length structure has been determined in complex with

substrates and its product[27–29]. To date, no full-length crystal structure of eukaryotic GFAT is available. However, structures of the isolated isomerase domains of *Candida albicans* GFAT (Gfa) and human GFAT-1 were reported[30–32]. Overall, the eukaryotic isomerase domains are very similar to the bacterial homolog. Moreover, the *C. albicans* Gfa isomerase domain was crystallized in the presence of the feedback inhibitor UDP-GlcNAc and revealed the UDP-GlcNAc binding site within the isomerase domain[31]. This binding site was confirmed in human GFAT-1[33]. Although UDP-GlcNAc binds to GFAT's isomerase domain, it inhibits the glutaminase function and thus GlcN6P production, suggesting interdomain communication[31,34]. Interfering with GFAT regulation might open an avenue to pharmacological modulation of the HP.

Here, we present the full-length human GFAT structure and delineate how single amino acid substitutions modulate GFAT activity. Structural and functional analyses of point mutants show that their gain-of-function results from loss of UDP-GlcNAc inhibition. Going beyond in vitro assays, we demonstrate the relevance of the GFAT gain-of-function substitution in regulating the HP in mammalian cells.

## Results

**Structure of full-length human GFAT-1.** To understand HP regulation at the molecular level, we determined the crystal structure of active full-length human GFAT-1. As N- or C-terminal tags interfere with GFAT-1 activity[35], we inserted an internal His₆-tag between Gly299 and Asp300 (Supplementary Fig. 1a), which does not interfere with GFAT-1 kinetic properties[36]. We established a protocol for large-scale production of active, internally His₆-tagged GFAT-1 using the MultiBac baculovirus expression system with subsequent purification via immobilized metal affinity chromatography and size-exclusion chromatography[37]. Tetragonal GFAT-1 crystals formed within a few days and diffracted to a resolution limit of 2.4 Å. Data collection and refinement statistics are given in Tables 1 and 2. Two GFAT-1 monomers were present in the asymmetric unit, which were termed monomer A and B according to the chain identifier in the PDB files. The complete structure was modeled into the electron density map except for two flexible loops of the glutaminase domain (residues 228–239 and 295–299) that include the internal His₆-tag. The two GFAT-1 monomers in the asymmetric unit form an asymmetric dimer through direct interactions of the isomerase domains while the glutaminase domains point outward to opposite sides (Fig. 1b).

**Structural comparison of human GFAT-1 with its homologs.** The human isomerase domain consists of two sugar isomerase (SIS) sub-domains: both are composed of five-stranded parallel β-sheets flanked by two or three α-helices on both sides (Fig. 1c). Overall, the isomerase domain is very similar to the respective structures from *E. coli*, *C. albicans*, and the previously published isolated human isomerase domain (Supplementary Fig. 1b).

The structure of the glutaminase domain of human GFAT-1 shows a typical N-terminal nucleophile (Ntn) hydrolase fold with two β-sheets composed of seven and five antiparallel β-strands sandwiched between two layers of three α-helices (αββα-core, Fig. 1c). Two short antiparallel β-strands (residues 221–223 (β9) and 303–305 (β14), Fig. 1c, Supplementary Fig. 1a), the latter originating from the linker between the two domains, cover one side of the αββα-core. The overall αββα-core of the human GFAT-1 glutaminase domain is similar to *E. coli* GlmS, while β-strands and loops connecting the α-helices and β-sheets are more extended in the human enzyme (Supplementary Fig. 1a, c). At least two phosphorylation sites, S235 and S243, are located within

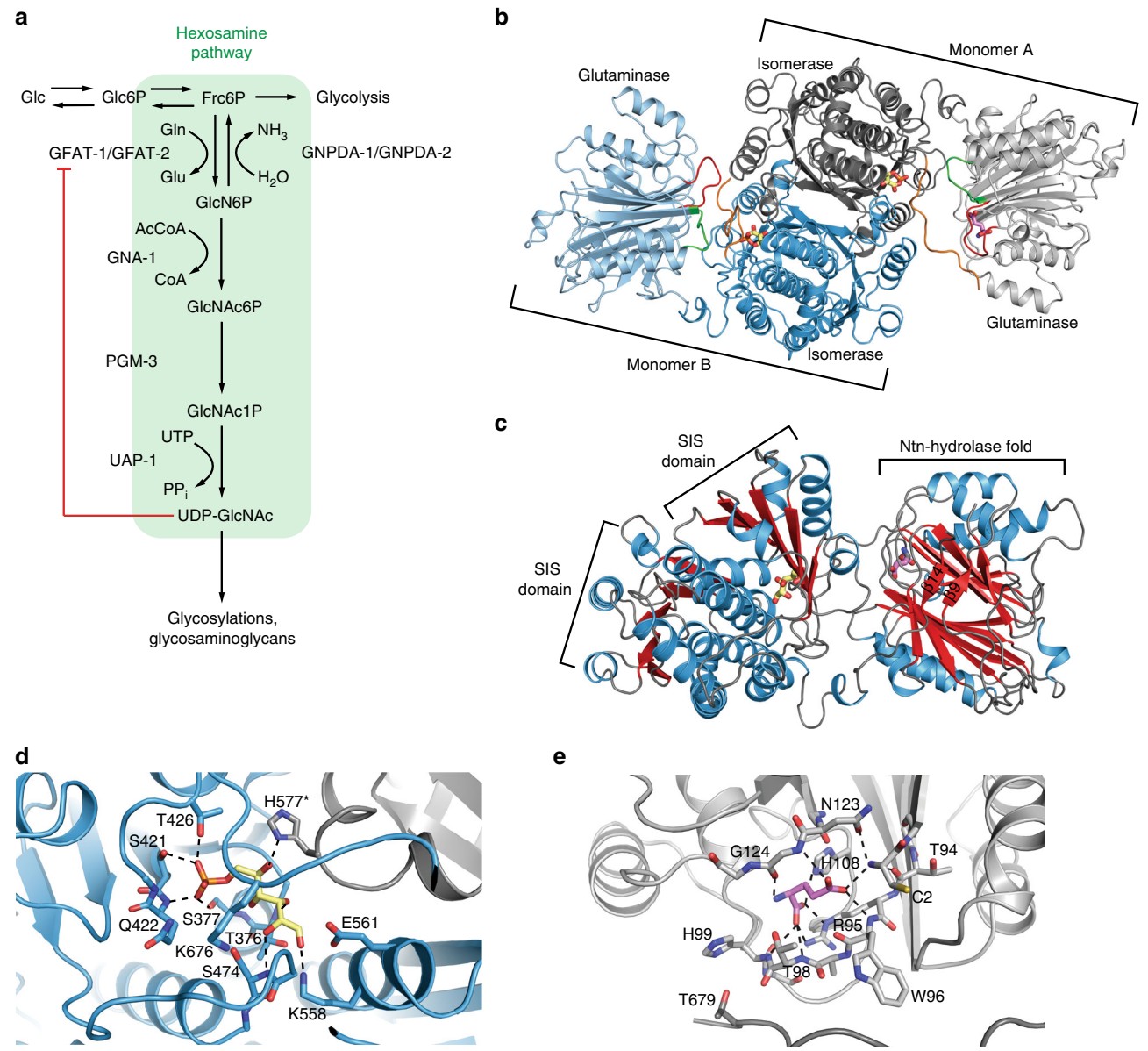

**Fig. 1 Structure of human GFAT-1, the key enzyme of the hexosamine pathway. a** Schematic representation of the hexosamine pathway (green box). The enzymes in the pathway are glutamine fructose-6-phosphate amidotransferase (GFAT-1/-2), glucosamine-6-phosphate *N*-acetyltransferase (GNA-1), phosphoglucomutase (PGM-3), UDP-*N*-acetylglucosamine pyrophosphorylase (UAP-1), and glucosamine-6-phosphate deaminase (GNPDA-1/−2). UDP-GlcNAc inhibits eukaryotic GFAT (red line). **b** Overall structure of the human GFAT-1 dimer in cartoon representation. The N-terminal glutaminase domains are colored in light blue and light gray, and the C-terminal isomerase domains in marine and dark gray. Glc6P (yellow sticks) and L-Glu (violet sticks) are highlighted, as well as important loops discussed in this manuscript: R-loop (green), Q-loop (red), and C-loop (orange). **c** Secondary structure elements of human GFAT-1. β-Sheets are colored in red and α-helices in blue. Glc6P (yellow sticks) and L-Glu (violet sticks) are highlighted. The isomerase domain (left) consists of two sugar isomerase (SIS) sub-domains. The glutaminase domain (right) is composed of a Ntn-hydrolase fold (αββα-core) with two short antiparallel β-sheets covering one side (β9 and β14). **d, e** Active sites of human GFAT-1. The protein is in cartoon representation; residues involved in substrate binding or catalysis are highlighted as sticks, and dashed lines indicate key interactions. **d** Frc6P-binding site formed by both isomerase domains. **e** L-Gln-binding site in one glutaminase domain.

these extended loops and S243 was found phosphorylated in both mass spectrometry analysis and the crystal structure (Supplementary Fig. 1a, d).

**GFAT-1 active sites are conserved from bacteria to humans.** GFAT-1 was crystallized in the presence of its substrate Frc6P and the product L-Glu. Corresponding electron density was found in both active sites. At the given resolution, however, we cannot unambiguously determine whether the substrate Frc6P or Glc6P, which is formed in the absence of L-Gln, was present in the

isomerase active site. As the presence of the linear sugar suggests isomerization activity and the equilibrium constant for Frc6P-Glc6P-isomerization favors Glc6P formation[38], we decided to model the product Glc6P in our crystal structures (Supplementary Fig. 1e). Residues of both isomerase domains of the GFAT-1 dimer contribute to each Glc6P-binding site. Thr376, Ser474, and Lys558 of one monomer, and His577* of the other monomer coordinate the hydroxyl groups of Glc6P through hydrogen bonding. Further hydrogen bonds are formed between the sugar phosphate group and Ser377, Ser421, Gln422, and Thr426

**Table 1 Data collection and refinement statistics of wild type GFAT-1.**

| | GFAT-1 WT +Glu+Glc6P | GFAT-1 WT +Glu+GlcN6P | GFAT-1 WT w/o Glu+Glc6P | GFAT-1 WT w/o Glu+Glc6P +UDPGlcNAc | GFAT-1 WT +Glu+Glc6P +UDPGlcNAc |
|---|---|---|---|---|---|
| Wavelength (Å) | 0.98 | 0.98 | 1.00 | 1.00 | 1.00 |
| Resolution range (Å) | 40.13–2.35 (2.44–2.35) | 48.33–2.33 (2.41–2.33) | 48.81–2.50 (2.59–2.50) | 46.36–2.50 (2.59–2.50) | 48.66–2.53 (2.62–2.53) |
| Space group | P 4₁ 2₁ 2 | P 4₁ 2₁ 2 | P 4₁ 2₁ 2 | P 4₁ 2₁ 2 | P 4₁ 2₁ 2 |
| $a, b, c$ (Å) | 153.9 153.9 166.3 | 152.8 152.8 165.4 | 153.0 153.0 167.9 | 152.4 152.4 169.3 | 152.6 152.6 166.5 |
| $\alpha, \beta, \gamma$ (°) | 90 90 90 | 90 90 90 | 90 90 90 | 90 90 90 | 90 90 90 |
| Total reflections | 1,068,061 (96,281) | 1,870,831 (170,057) | 891,471 (74,962) | 685,152 (65,470) | 866,824 (78,008) |
| Unique reflections | 82,721 (7933) | 84,017 (8181) | 69,161 (6763) | 69,149 (6736) | 65,754 (6299) |
| Multiplicity | 12.9 (12.1) | 22.3 (20.8) | 12.9 (11.1) | 9.9 (9.7) | 13.2 (12.4) |
| Completeness (%) | 99.6 (96.8) | 99.8 (98.8) | 99.9 (98.9) | 99.8 (98.9) | 99.7 (97.0) |
| Mean $I$/sigma ($I$) | 18.10 (1.57) | 18.80 (1.08) | 20.61 (1.11) | 19.93 (1.39) | 14.98 (1.05) |
| Wilson B-factor | 61.6 | 60.1 | 62.8 | 58.8 | 57.5 |
| $R_{merge}$ (%) | 8.4 (133.7) | 11.8 (230.2) | 10.1 (189.9) | 9.6 (146.2) | 16.0 (215.9) |
| $R_{meas}$ (%) | 8.8 (139.5) | 12.0 (235.9) | 10.5 (199.1) | 10.1 (154.3) | 16.7 (225.1) |
| $R_{pim}$ (%) | 2.4 (39.6) | 2.5 (51.1) | 2.9 (59.3) | 3.2 (48.7) | 4.6 (63.1) |
| $CC_{1/2}$ (%) | 99.9 (68.6) | 99.9 (55.8) | 99.9 (45.2) | 99.9 (55.6) | 99.9 (41.7) |
| $CC^*$ (%) | 100 (90.2) | 100 (84.6) | 100 (78.9) | 100 (84.5) | 100 (76.7) |
| Reflections used in refinement | 82,715 (7933) | 83,998 (8181) | 69,156 (6763) | 69,139 (6736) | 65,747 (6299) |
| Reflections used for $R$-free | 1991 (193) | 1927 (185) | 1936 (189) | 1935 (189) | 1972 (190) |
| $R_{work}$ (%) | 17.5 (25.8) | 19.0 (31.7) | 19.4 (31.4) | 19.0 (30.7) | 19.7 (30.4) |
| $R_{free}$ (%) | 20.9 (31.5) | 22.5 (36.9) | 21.4 (32.9) | 21.7 (36.0) | 22.4 (31.6) |
| $CC_{work}$ (%) | 96.9 (81.8) | 96.7 (72.8) | 95.9 (68.6) | 96.4 (71.5) | 96.0 (67.3) |
| $CC_{free}$ (%) | 93.2 (71.1) | 94.2 (62.4) | 96.6 (70.9) | 95.5 (59.4) | 94.1 (60.4) |
| Number of non-hydrogen atoms | 10,552 | 10,574 | 10,339 | 10,481 | 10,514 |
| Macromolecules | 10,440 | 10,440 | 10,236 | 10,296 | 10,332 |
| Ligands | 32 | 32 | 32 | 112 | 112 |
| Solvent | 80 | 102 | 71 | 73 | 70 |
| Protein residues | 1323 | 1321 | 1297 | 1304 | 1306 |
| RMS (bonds) (Å) | 0.002 | 0.002 | 0.003 | 0.002 | 0.002 |
| RMS (angles) (°) | 0.45 | 0.46 | 0.49 | 0.45 | 0.44 |
| Ramachandran favored (%) | 97 | 96 | 96 | 96 | 96 |
| Ramachandran allowed (%) | 3.2 | 3.5 | 4.1 | 3.6 | 3.9 |
| Ramachandran outliers (%) | 0.076 | 0.15 | 0.23 | 0.15 | 0.15 |
| Rotamer outliers (%) | 0.17 | 0.17 | 0.27 | 0.26 | 0.088 |
| Clashscore | 0.57 | 0.72 | 0.73 | 0.67 | 0.62 |
| Average B-factor | 90.33 | 89.70 | 103.10 | 90.04 | 92.39 |
| Macromolecules | 90.68 | 90.13 | 103.53 | 90.58 | 92.99 |
| Ligands | 60.92 | 57.45 | 70.99 | 66.57 | 64.29 |
| Solvent | 57.06 | 55.88 | 54.91 | 49.71 | 49.12 |
| Number of TLS groups | 4 | 4 | 4 | 4 | 4 |
| PDB code | 6R4E | 6SVO | 6R4F | 6R4G | 6SVP |

Statistics for the highest-resolution shell are shown in parentheses

(Fig. 1d, Supplementary Fig. 1f). The catalytically relevant residues Lys558, Glu561, His577*, and Lys676 are conserved from *E. coli* to humans (Supplementary Fig. 1a)[39].

Mass spectrometry analysis of the glutaminase domain revealed that the initial methionine is removed, resulting in a free α-amino group of the catalytic Cys2, as published previously[27]. The product L-Glu is bound to the glutaminase active site in one monomer (monomer A), but is absent in the second monomer (monomer B). In the L-Glu-free state, Cys2 faces the substrate-binding pocket, while it points away from the active site in the presence of L-Glu (Supplementary Fig. 1g). In the L-Glu-bound monomer, the α-amino group of Cys2 forms hydrogen bonds to Thr94, Asn123, and to the δ-carboxyl-group of L-Glu. The L-Glu binding pocket itself is further formed by Arg95, Trp96, Thr98, His99, His108, and Gly124 (Fig. 1e, Supplementary Fig. 1h, i). Comparison of the glutaminase domains with the *E. coli* structure revealed that all relevant residues involved in glutamine hydrolysis (Cys2, Asn123, Gly124, and Thr679) are fully conserved (Supplementary Fig. 1a)[17,27].

Thus, both binding pockets and the catalytic residues are evolutionary conserved from *E. coli* to human GFAT-1, suggesting a similar reaction mechanism as described in bacteria[40].

**GFAT-1 forms an asymmetric dimer**. The two crystallographically independent GFAT-1 monomers form a dimer. The dimer is asymmetric because the two glutaminase domains are oriented differently relative to their respective isomerase moieties, which form a symmetric assembly (Figs. 1b and 2a). The position of the glutaminase domain of monomer A is stabilized by crystal contacts. In contrast, the glutaminase domain of monomer B makes no crystal contacts and is more flexible resulting in higher B-factors (Supplementary Fig. 2a) and partially low electron density (Supplementary Fig. 2b). Due to the conformational shift of the glutaminase domains relative to their isomerase domains, the cleft between the two domains is more open in monomer A than in monomer B. This open conformation in monomer A allows conformational changes at the active sites with loop movements, which do not occur in the closed conformation in

**Table 2 Data collection and refinement statistics of wild type and point mutant GFAT-1.**

| | GFAT-1 WT +Glu+Glc6P +UDPGalNAc | GFAT-1 G461E | GFAT-1 G461E +UDPGlcNAc | GFAT-1 G451E | GFAT-1 G451E +UDP-GlcNAc |
|---|---|---|---|---|---|
| Wavelength (Å) | 1.00 | 0.97 | 0.97 | 1.00 | 1.00 |
| Resolution range (Å) | 49.15–2.48 (2.57–2.48) | 48.68–2.59 (2.68–2.59) | 48.21–2.72 (2.81–2.72) | 48.98–2.24 (2.32–2.24) | 48.73–2.42 (2.51–2.42) |
| Space group | P 4₁ 2₁ 2 | P 4₁ 2₁ 2 | P 4₁ 2₁ 2 | P 4₁ 2₁ 2 | P 4₁ 2₁ 2 |
| a, b, c (Å) | 152.0 152.0 165.8 | 152.8 152.8 166.0 | 152.5 152.5 164.9 | 154.1 154.1 162.9 | 153.2 153.2 162.5 |
| $\alpha, \beta, \gamma$ (°) | 90 90 90 | 90 90 90 | 90 90 90 | 90 90 90 | 90 90 90 |
| Total reflections | 601,542 (57,351) | 613,756 (59,726) | 464,957 (46,298) | 690,080 (646,13) | 992,398 (91,739) |
| Unique reflections | 68,982 (6701) | 61,581 (5916) | 52,752 (5146) | 93,589 (9028) | 74,011 (7226) |
| Multiplicity | 8.7 (8.6) | 10.0 (10.1) | 8.8 (9.0) | 7.4 (7.2) | 13.4 (12.7) |
| Completeness (%) | 99.8 (98.3) | 99.6 (97.2) | 99.7 (98.9) | 99.7 (97.5) | 99.9 (99.0) |
| Mean $I/\text{sigma}$ ($I$) | 14.75 (1.15) | 17.49 (1.41) | 16.86 (1.44) | 16.87 (1.54) | 18.36 (1.54) |
| Wilson B-factor | 57.5 | 65.0 | 71.5 | 45.7 | 49.7 |
| $R_{merge}$ (%) | 11.0 (157.4) | 9.7 (144.5) | 10.0 (142.9) | 8.0 (109.8) | 12.9 (163.2) |
| $R_{meas}$ (%) | 11.7 (167.4) | 10.2 (152.2) | 10.6 (151.7) | 8.6 (118.3) | 13.4 (170.0) |
| $R_{pim}$ (%) | 3.9 (56.1) | 3.2 (47.0) | 3.5 (50.2) | 3.2 (43.5) | 3.6 (47.4) |
| $CC_{1/2}$ (%) | 99.9 (49.6) | 99.9 (54.1) | 99.9 (54.9) | 99.9 (62.7) | 99.9 (59.9) |
| CC* (%) | 100 (81.5) | 100 (83.8) | 100 (84.2) | 100 (87.8) | 100 (86.5) |
| Reflections used in refinement | 68,971 (6701) | 61,566 (5916) | 52,734 (5145) | 93,581 (9029) | 74,006 (7226) |
| Reflections used for R-free | 1935 (177) | 1967 (181) | 1940 (209) | 1976 (192) | 1991 (195) |
| $R_{work}$ (%) | 18.9 (29.3) | 19.2 (29.2) | 20.5 (30.6) | 18.8 (27.1) | 19.4 (27.9) |
| $R_{free}$ (%) | 22.2 (33.5) | 21.9 (31.1) | 23.4 (31.6) | 20.4 (31.8) | 22.3 (30.9) |
| $CC_{work}$ (%) | 96.1 (71.8) | 96.2 (73.5) | 95.2 (70.1) | 96.5 (78.3) | 95.9 (77.8) |
| $CC_{free}$ (%) | 94.5 (47.9) | 93.9 (66.5) | 93.6 (68.9) | 96.0 (70.9) | 94.9 (79.8) |
| Number of non-hydrogen atoms | 10,565 | 10,463 | 10,270 | 10,617 | 10,570 |
| Macromolecules | 10,390 | 10,367 | 10,228 | 10,454 | 10,332 |
| Ligands | 112 | 32 | 32 | 32 | 112 |
| Solvent | 63 | 64 | 10 | 131 | 126 |
| Protein residues | 1316 | 1310 | 1292 | 1320 | 1304 |
| RMS (bonds) (Å) | 0.003 | 0.002 | 0.002 | 0.003 | 0.002 |
| RMS (angles) (°) | 0.48 | 0.45 | 0.44 | 0.53 | 0.44 |
| Ramachandran favored (%) | 96 | 96 | 96 | 98 | 97 |
| Ramachandran allowed (%) | 3.8 | 3.7 | 4.1 | 2 | 2.4 |
| Ramachandran outliers (%) | 0.077 | 0.15 | 0 | 0.15 | 0.23 |
| Rotamer outliers (%) | 0.087 | 0.17 | 0.089 | 0.17 | 0.088 |
| Clashscore | 0.48 | 0.72 | 0.83 | 0.67 | 0.77 |
| Average B-factor | 87.35 | 109.55 | 115.46 | 70.93 | 78.97 |
| Macromolecules | 87.83 | 109.98 | 115.69 | 71.37 | 79.63 |
| Ligands | 64.10 | 78.68 | 62.07 | 44.41 | 54.72 |
| Solvent | 49.35 | 56.18 | 53.11 | 42.56 | 46.88 |
| Number of TLS groups | 4 | 4 | 4 | 4 | 4 |
| PDB code | 6SVM | 6R4I | 6SVQ | 6R4H | 6R4J |

Statistics for the highest-resolution shell are shown in parentheses

monomer B. L-Glu was found bound to the glutaminase domain of the open monomer A, while in monomer B no L-Glu was detected (Fig. 1b). In monomer B, Glc6P binds to the isomerase domain and the nine C-terminal residues, the so-called C-loop (residues 670–681), cover the sugar-binding pocket (Fig. 2b, Supplementary Fig. 2c). The C-loop is stabilized by interactions with the side chains of Tyr35 and Trp96 of the glutaminase domain and by hydrogen bonds between Tyr32 and Arg33 with the C-loop backbone (Fig. 2b, Supplementary Fig. 2c). Compared to the *E. coli* structures, the conformation of monomer B is consistent with the first steps of the catalytic cycle, comprising initial sugar binding and structural ordering of C-loop and glutaminase domain[40]. In monomer A, isomerase and glutaminase domains are both occupied by Glc6P and L-Glu, respectively. Here, in the presence of the product L-Glu, the Q-loop (residues 95–102) covers the glutaminase site, as observed in the isolated glutaminase domain from *E. coli*[27]. In addition, there is a conformational change of the neighboring R-loop (residues 31–36) (Fig. 2b, c). L-Glu-induced shifts of the Q- and R-loops trigger side chain rotations of Tyr35 (R-loop) and Trp96 (Q-loop),

destabilizing the C-loop (Fig. 2b, d, Supplementary Fig. 2d). Furthermore, the hydrogen bonds between Tyr32 and Arg33 and the C-loop are disrupted in the presence of L-Glu. Together, these structural differences increase the C-loop's flexibility, indicated by higher B-factors and poor electron density in monomer A (Supplementary Fig. 2a, e). Thr679 of the C-loop is involved in L-Gln hydrolysis. Consistently, the C-loop is in close proximity to the Q-loop when the latter shields the active site (monomer A) (Fig. 2b, Supplementary Fig. 2d, e). Although both active sites are occupied by ligands in monomer A, no ammonia channel connecting the glutaminase and the isomerase site is formed. These data suggest that the Q-loop covers the glutaminase active site upon L-Glu binding, but as it is the product, it cannot induce the coupling of both active sites by ammonia channel formation. We wondered how the presence of L-Glu and GlcN6P would affect the structure and co-crystallized GFAT-1 in the presence of both product molecules. The GlcN6P-bound structure formed the same asymmetric dimer as the Glc6P-bound structure and a linear sugar was present in the active site (Supplementary Fig. 2f, g, h). Compared to human GFAT-1, the GlcN6P-bound *E. coli*

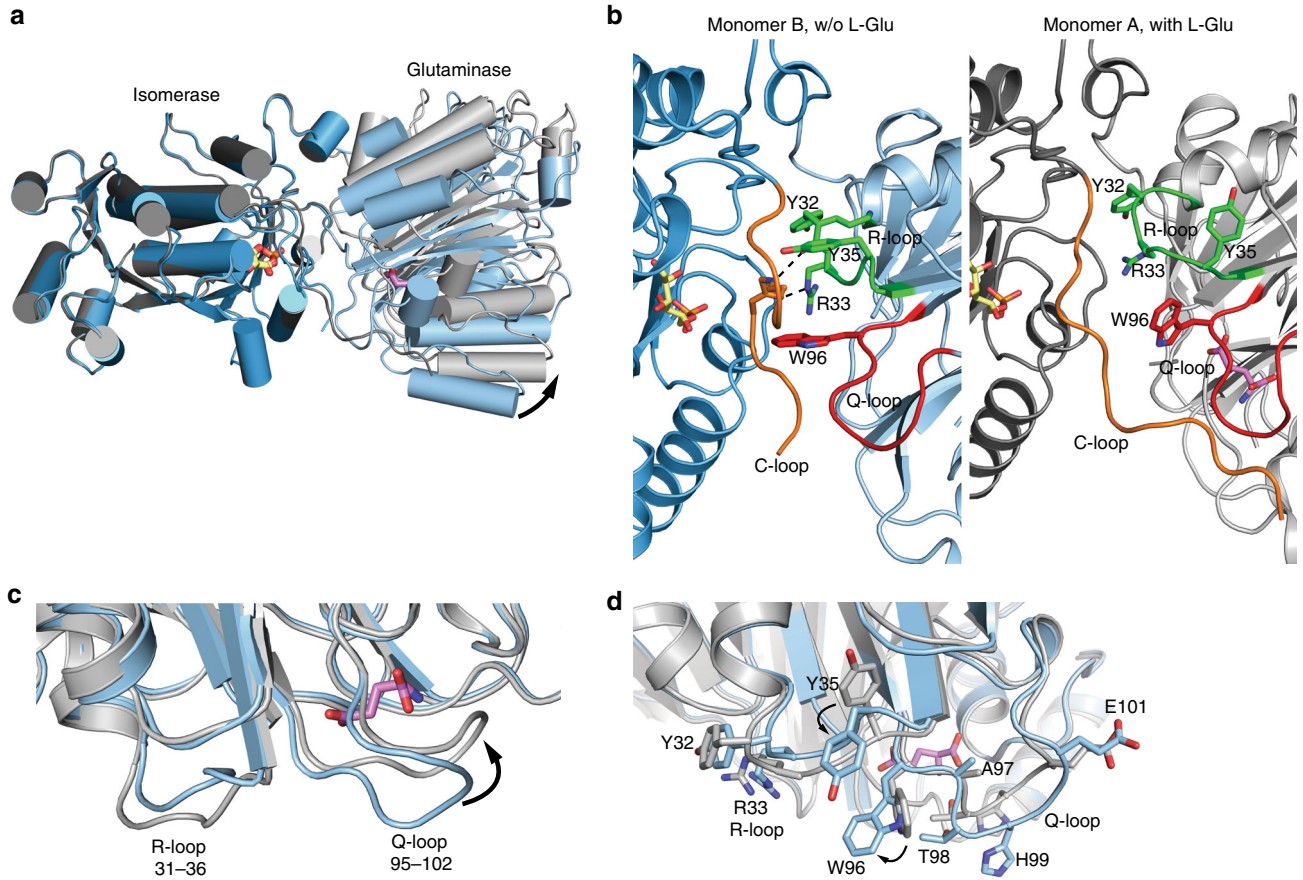

**Fig. 2 GFAT-1 forms an asymmetric dimer. a–d** Analysis of structural differences between human GFAT-1 monomers (monomer A and monomer B). Proteins are presented as cartoons. Monomer A is colored in dark gray (isomerase domain) and light gray (glutaminase domain), while monomer B is marine (isomerase domain) and light blue (glutaminase domain). Glc6P (yellow sticks) and L-Glu (violet sticks) are highlighted. **a** Superposition of isomerase domains of monomer A and monomer B. α-Helices are presented as cylinders. **b** Comparison of the C-loop orientation in monomer B (left) and monomer A (right). Residues interacting with the C-loop in chain B are highlighted (sticks), as well as important loops: R-loop (green), Q-loop (red), and C-loop (orange). Black lines indicate hydrogen bonds. **c** Superposition of glutaminase domains of monomer A and monomer B reveal structural differences of the Q- and R-loop. **d** Close-up view on the structural rearrangements in the Q- and R-loop in a superposition of the glutaminase domains of monomer A and monomer B.

structure shows the pyranose ring GlcN6P in the active site and no electron density for the glutaminase domain is observed, although the full-length protein was crystallized[41]. While in the *E. coli* structure the product might have already left the active site, destabilizing the glutaminase domain, our structure represents the last step of catalysis, just before departure of the GlcN6P product.

**Structural alterations in the absence of glutamate.** Given that the occupancy of the glutaminase site by L-Glu is linked to the conformation of the Q- and R-loops, we wanted to know the conformation of these loops in the absence of L-Glu. Since crystallization without glutamate yielded poorly diffracting crystals, we crystallized GFAT-1 in the presence of glutamate and subsequently removed it in several dilution steps.

The removal of glutamate, as indicated by the electron density (Supplementary Fig. 3a), leads to major changes in the glutaminase domain of the previously L-Glu-bound monomer A (Fig. 3a): first, lower electron density and higher B-factors of the Q-loop indicate higher flexibility and a destabilization of this loop at the glutaminase site (Fig. 3b, Supplementary Fig. 3b). This is consistent with the observation that residues of the Q-loop stabilize glutamate binding. Second, the R-loop changes orientation with large side chain movements of Arg33 and Tyr35

(Fig. 3a, c). In the presence of glutamate, Arg33 points to the cleft between both domains and forms a salt bridge with Asp262 of the glutaminase domain. In the absence of glutamate, however, Arg33 is rotated towards the isomerase domain and forms a salt bridge with Asp667 of the isomerase domain. The Arg33 movement frees space between Tyr32 and the catalytically active Cys2, which is filled by Tyr35 (Fig. 3c). Third, in the absence of glutamate, the C-loop could not be modeled due to the lack of electron density indicating its high flexibility (Fig. 3a, b). However, the removal of glutamate did not affect the conformation of monomer B. Together, the glutamate-free structure of monomer A suggests a communication of the glutaminase site through Arg33 with the isomerase domain.

**Interaction of UDP-GlcNAc and UDP-GalNAc with human GFAT-1.** The main distinctive feature between prokaryotic and eukaryotic GFAT is the feedback inhibition by the HP product UDP-GlcNAc, which inhibits the glutaminase function[34]. To mechanistically understand GFAT-1 feedback inhibition, we next analyzed UDP-GlcNAc interaction with GFAT-1. Co-crystallization of GFAT-1 with UDP-GlcNAc did not yield well-diffracting crystals. However, we successfully soaked UDP-GlcNAc into the human GFAT-1 crystals and analyzed the crystal structure. UDP-GlcNAc was bound to both GFAT-1 monomers

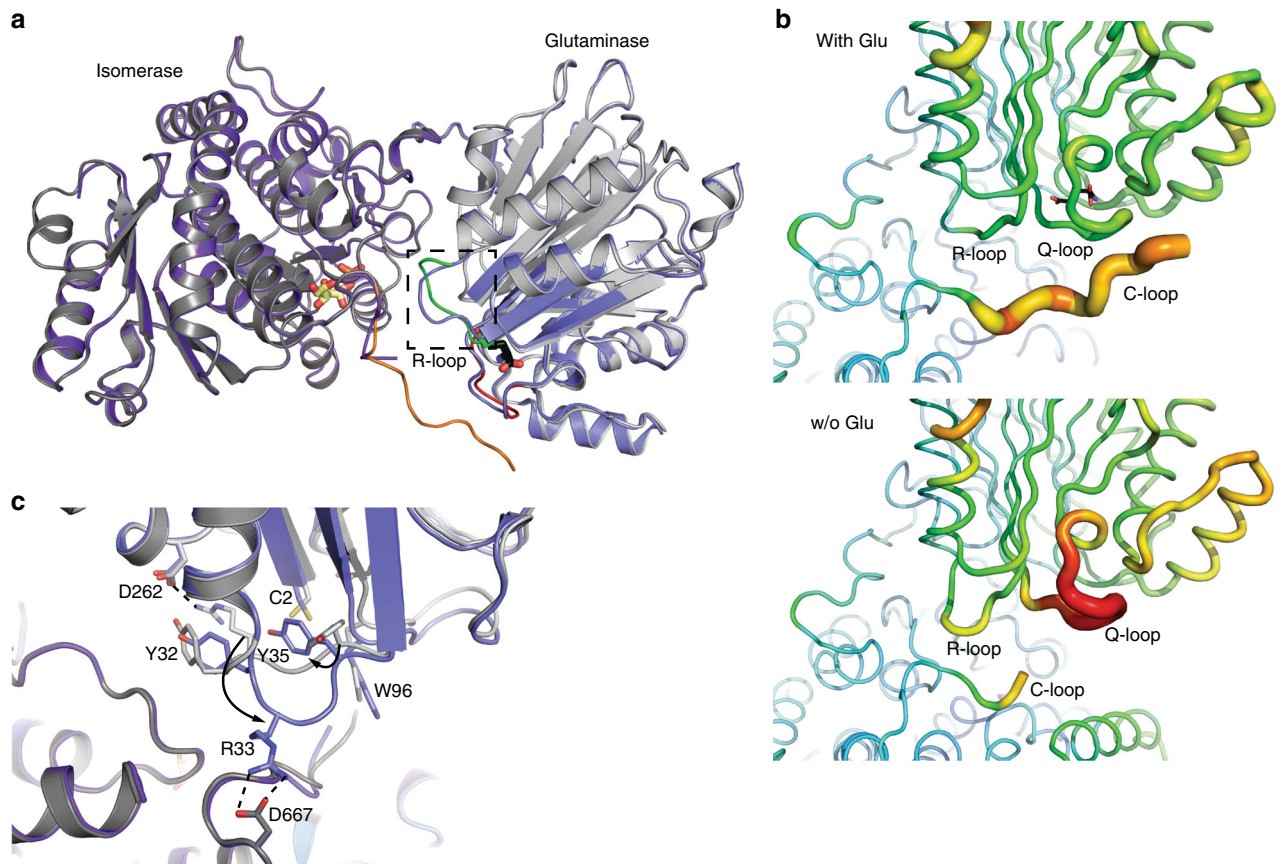

**Fig. 3 Glutamate determines conformation of Q-loop, R-loop, and C-loop. a, c** Superposition of Glu-free and Glu-bound GFAT-1 monomer A. Proteins are presented as cartoons. Glu-bound GFAT-1 is depicted in gray and Glu-free GFAT-1 is colored in purple. Glc6P (yellow sticks) and L-Glu (black sticks) are highlighted, as well as important loops: R-loop (green), Q-loop (red), and C-loop (orange). **a** Overview. Glutamate presence/absence influences the conformation of the R-loop (black box). **b** Representation of B-factors in Glu-bound (upper) and Glu-free (lower) wild type human GFAT-1 as putty cartoon. Colored from low to high values (20–200 Å$^2$, blue to red). L-Glu (black sticks) is highlighted. Glutamate absence destabilizes the Q-loop. **c** Glutamate absence induces R-loop flipping with large movements of side chains of R33 and Y35.

in the asymmetric unit. As shown previously, UDP-GlcNAc binds with a cation to the GFAT-1 isomerase domain (Fig. 4a–c, Supplementary Fig. 4a, b)[31]. UDP-GlcNAc binding is stabilized by hydrogen bonds to Gly355, Thr447, Thr458, Gly461, as well as ionic interactions between the pyrophosphates and Arg343 and His463 (Fig. 4c, Supplementary Fig. 4b). Additionally, we identified hydrogen bonds between UDP-GlcNAc and Gln310 of the interdomain linker in monomer A (Fig. 4c, Supplementary Fig. 4b). These were not observed in monomer B due to an increased distance caused by the shift of the glutaminase domain relative to the isomerase domain (Fig. 2a) and poor electron density of Gln310. Binding of UDP-GlcNAc induced local side chain reorientations of His463 and Asn465 in both monomers (Fig. 4d). While the UDP moiety of UDP-GlcNAc interacts with the isomerase domain through multiple amino acid residues, there are only two interactions of the N-acetylglucosamine moiety with the protein (Fig. 4c, Supplementary Fig. 4b). To test the role of the sugar in GFAT-1 inhibition, we used the UDP-GlcNAc epimer uridine 5′-diphospho-N-acetyl-D-galactosamine (UDP-GalNAc), which differs only in the orientation of the hydroxyl group at C4 of the sugar ring that is pointing towards the interdomain linker. UDP-GalNAc was detected in the UDP-GlcNAc binding site after crystal soaking and induced the same side chain movements (Fig. 4e, Supplementary Fig. 4c–e). To investigate its role in GFAT-1 inhibition, we next analyzed GFAT-1 inhibition in activity assays.

**GFAT-1 activity and feedback inhibition**. To analyze the activity of GFAT-1 in vitro, we used activity assays for each domain: L-Glu production by the glutaminase domain was measured by a coupled activity assay with glutamate dehydrogenase (GDH). D-GlcN6P synthesis of the isomerase domain was monitored using a coupled activity assay including glucosamine-6-phosphate N-acetyltransferase (GNA-1), the second enzyme of the HP (Supplementary Fig. 4f). In kinetic measurements, we observed both L-Gln hydrolysis and D-GlcN6P synthesis (Table 3, Fig. 4f, Supplementary Fig. 4g). These numbers are in accordance with previous studies[42]. The GDH-coupled glutaminase domain activity assay was used to characterize UDP-GlcNAc and UDP-GalNAc effects. As a negative control we generated a GFAT-1 variant that prevents UDP-GlcNAc binding by mutating Gly461, which is located at the bottom of the UDP-GlcNAc-binding pocket (Fig. 4c). Indeed, UDP-GlcNAc did not bind to the G461E mutant of GFAT-1 after crystal soaking (Supplementary Fig. 4h). Kinetic measurements showed that GFAT-1 G461E has similar kinetic properties like wild type GFAT-1 for L-Glu and D-GlcN6P synthesis (Table 3, Supplementary Fig. 4g, i). In vitro, wild type human GFAT-1 was inhibited by UDP-GlcNAc in a dose-dependent manner with an IC$_{50}$ of 43.3 (−2.3/+2.5) μM (Fig. 4g). GFAT-1 G461E did not respond to UDP-GlcNAc treatment, as expected from the crystal structure (Supplementary Fig. 4j). Of note, in wild type GFAT1, UDP-GalNAc showed an inhibitory effect only at much higher doses compared to UDP-GlcNAc

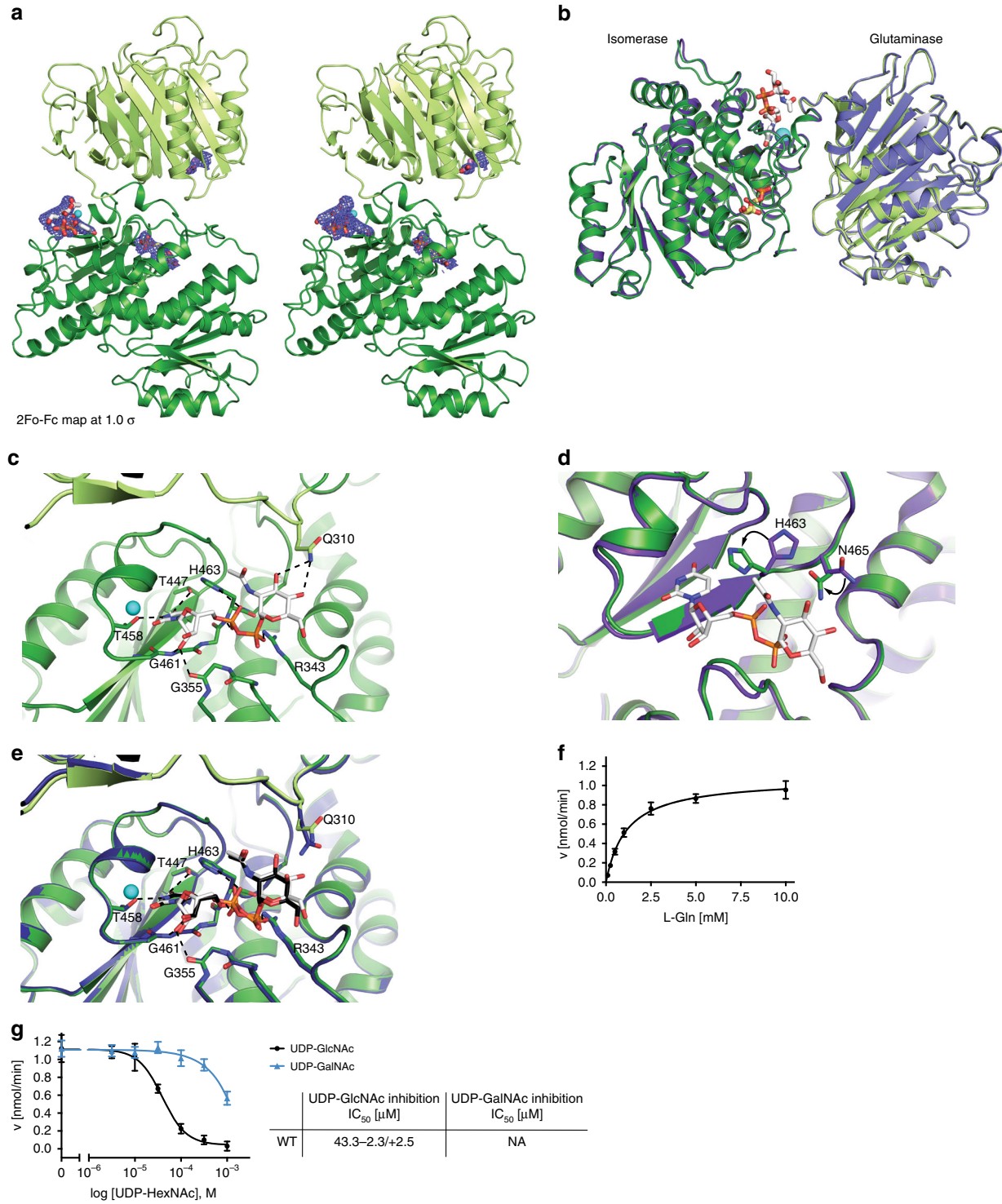

(Fig. 4g). Taken together, GFAT-1 was responsive to UDP-GlcNAc and feedback inhibition of UDP-GlcNAc involves a critical interaction between the sugar's C4 hydroxyl group and the interdomain linker.

**Inhibition of regulatory feedback by a single point mutation.**
Having solved the structure of GFAT-1, we aimed to understand

the mechanism of GFAT-1 activation through a single amino acid substitution (G451E) that elevated HP flux and extended lifespan in *C. elegans*[4]. Positioned in an evolutionary conserved region of the isomerase domain, Gly451 is in close proximity to the UDP-GlcNAc-binding site, suggesting that the mutation might interfere with UDP-GlcNAc inhibition (Supplementary Fig. 1a, 5a). Crystal structure analysis of GFAT-1 G451E revealed no major structural changes compared to wild type GFAT-1 (Fig. 5a, b).

Fig. 4 GFAT-1 inhibition by UDP-GlcNAc and UDP-GalNAc. a–d Structural analysis of UDP-GlcNAc binding to wild type-GFAT-1. Proteins are presented as cartoons. UDP-GlcNAc-bound GFAT-1 is depicted in green and UDP-GlcNAc-free GFAT-1 is colored in purple. Glc6P (yellow sticks), L-Glu (violet sticks), UDP-GlcNAc (white sticks), and $Mg^{2+}$ (cyan sphere) are highlighted. a Stereo image of GFAT-1 monomer A. The 2Fo−Fc maps (blue) of the ligands have a contour level of 1.0 RMSD. b Superposition of UDP-GlcNAc-free and UDP-GlcNAc-bound GFAT-1. c UDP-GlcNAc binding site at the isomerase domain. Binding is mediated by residues of the isomerase domain, as well as interactions with Gln310 from the interdomain linker region. Residues involved in UDP-GlcNAc binding are highlighted as sticks, and dashed lines indicate the most important interactions. d Locally occurring side chain movements upon UDP-GlcNAc binding. e Superposition of UDP-GlcNAc-binding site at the GFAT-1 isomerase domain in the presence of UDP-GlcNAc (green structure) or UDP-GalNAc (blue structure). UDP-GlcNAc (white sticks), UDP-GalNAc (black sticks), and $Mg^{2+}$ (cyan sphere) are highlighted. f L-Gln kinetic of wild type (WT, black circle) GFAT-1 (mean ± SEM, n = 5). g Representative UDP-GlcNAc (black circle) and UDP-GalNAc (blue triangle) inhibition of wild type GFAT-1 (mean ± SD, n = 5). Table: $IC_{50}$ UDP-GlcNAc values (mean ± SEM, n = 3). Source data are provided as a Source Data file.

**Table 3 Kinetic parameters.**

| | L-Glu production | | | D-GlcN6P production | | |
|---|---|---|---|---|---|---|
| | $K_m$ L-Gln [mM] | $k_{cat}$ [s$^{-1}$] | $k_{cat}/K_m$ [mM$^{-1}$ s$^{-1}$] | $K_m$ Frc6P [mM] | $k_{cat}$ [s$^{-1}$] | $k_{cat}/K_m$ [mM$^{-1}$ s$^{-1}$] |
| Wild type | 1.1 ± 0.2 | 3.6 ± 0.2 | 3.3 | 0.08 ± 0.01 | 1.7 ± 0.1 | 21.3 |
| G461E | 0.5 ± 0.1 | 3.1 ± 0.1 | 6.2 | 0.09 ± 0.01 | 2.1 ± 0.1 | 23.3 |
| G451E | 1.3 ± 0.2 | 2.2 ± 0.1 | 1.7 | 0.04 ± 0.01 | 0.8 ± 0.04 | 20.0 |
| Q307A | 0.6 ± 0.04 | 2.6 ± 0.04 | 4.3 | 0.15 ± 0.01 | 2.1 ± 0.1 | 14.0 |
| Q307A/G451E | 1.0 ± 0.05 | 2.1 ± 0.03 | 2.1 | 0.05 ± 0.01 | 1.1 ± 0.04 | 22.0 |

Wild type, G461E, and G451E: mean ± SEM, n = 5; Q307A and Q307A/G451E: mean ± SEM, n = 3

Notably, Glu451 forms a new hydrogen bond to Gln307 of the interdomain linker in the mutant structure (Fig. 5b). Analysis of the kinetic properties of the G451E mutant revealed a comparable $K_m$ for L-Gln-hydrolysis but had a reduced $k_{cat}$ compared to the wild type enzyme (Table 3, Fig. 5c). Furthermore, the $k_{cat}/K_M$ ratio for D-GlcN6P synthesis is comparable to the wild type, although both $K_m$ and $k_{cat}$ are smaller (Table 3, Supplementary Fig. 5b). Thus, although the L-Glu production rate was reduced, synthesis of D-GlcN6P was as efficient as in the wild type enzyme. Very remarkably, however, GFAT-1 G451E showed a drastically reduced sensitivity to UDP-GlcNAc feedback inhibition (Fig. 5d, Supplementary Fig. 5c). The analysis of GFAT-1 G451E crystals soaked with 5 mM UDP-GlcNAc revealed that this GFAT-1 variant was still competent to bind to UDP-GlcNAc (Fig. 5e, Supplementary Fig. 5d) at the high concentrations employed in this experiment. UDP-GlcNAc also induced the local side chain movements described above for the wild type within the binding pocket (Fig. 5f). To further analyze the role of the additional hydrogen bond between G451E and Gln307 of the interdomain linker, we characterized the mutant Q307A and the double-mutant Q307A/G451E. Q307A showed similar kinetic properties as wild type, while the kinetic parameters of Q307A/G451E were similar to G451E (Table 3, Supplementary Fig. 5e, f). Unexpectedly, Q307A showed a higher sensitivity to UDP-GlcNAc compared to wild type (Fig. 5d). In contrast to the G451E variant, the double mutant Q307A/G451E was inhibited at a lower UDP-GlcNAc dose (Fig. 5d). We conclude that GFAT-1 gain-of-function in the G451E variant results from a loss of regulation by UDP-GlcNAc-mediated feedback inhibition. Moreover, the increased UDP-GlcNAc sensitivity of the interdomain linker mutant Q307A again points to a critical role of the interdomain linker in UDP-GlcNAc inhibition.

**Evolutionary conservation of GFAT-1**. As UDP-GlcNAc inhibition is a feature of eukaryotic GFAT-1, we analyzed the evolutionary conservation of the interdomain linker from bacteria to higher organisms. To this end, we generated a sequence alignment comparing eukaryotic and prokaryotic GFATs and used the ConSurf server to highlight conserved regions within the

structure[43]. Quite unsurprisingly, both isomerase and glutaminase active sites are fully conserved between pro- and eukaryotes (Fig. 6a). In contrast, the interdomain linker showed a high heterogeneity within the prokaryotes, while the eukaryotic interdomain linker was well conserved (Fig. 6b). This further supports a key role of the interdomain linker in UDP-GlcNAc-mediated GFAT-1 inhibition.

**Loss of feedback inhibition activates GFAT-1 in vivo**. Understanding GFAT-1 gain-of-function by a specific point mutation, we next assessed the relevance of this mutation in mammalian cells. For this, we introduced the GFAT-1 G451E substitution in N2a mouse neuroblastoma cells by editing the endogenous locus using CRISPR/Cas9. Two independent cell lines carrying the homozygous GFAT-1 mutation were generated (Fig. 7a, Supplementary Fig. 6a). Given that HP activation confers strong tunicamycin resistance in C. elegans, we assessed cell survival upon tunicamycin treatment in the engineered N2a cells. Compared to wild type control cells, both GFAT-1 G451E lines were resistant to tunicamycin (Fig. 7b, Supplementary Fig. 6b). Further, we measured absolute levels of UDP-GlcNAc and UDP-GalNAc (together UDP-HexNAc) from cellular extracts by liquid chromatography coupled mass spectrometry. Both GFAT-1 G451E lines showed markedly increased steady-state levels of UDP-HexNAc, compared to the wild type control line (Fig. 7c, Supplementary Fig. 6c). To rule out that elevated UDP-GlcNAc levels were a consequence of increased GFAT-1 expression in the mutant cells, we quantified mRNA and protein. We found that GFAT-1 expression was decreased upon the G451E mutation (Fig. 7d–f, Supplementary Fig. 6d–f), indicating that the gain-of-function resulted from constant GFAT-1 activity due to the lack of feedback inhibition in vivo.

**Discussion**
Here we present the full-length crystal structure of human GFAT-1 and present mechanistic insights into its regulation that affects protein homeostasis though the HP. We performed soaking experiments with the physiological inhibitor UDP-GlcNAc and its epimer UDP-GalNAc. Both bind to the GFAT-1 isomerase

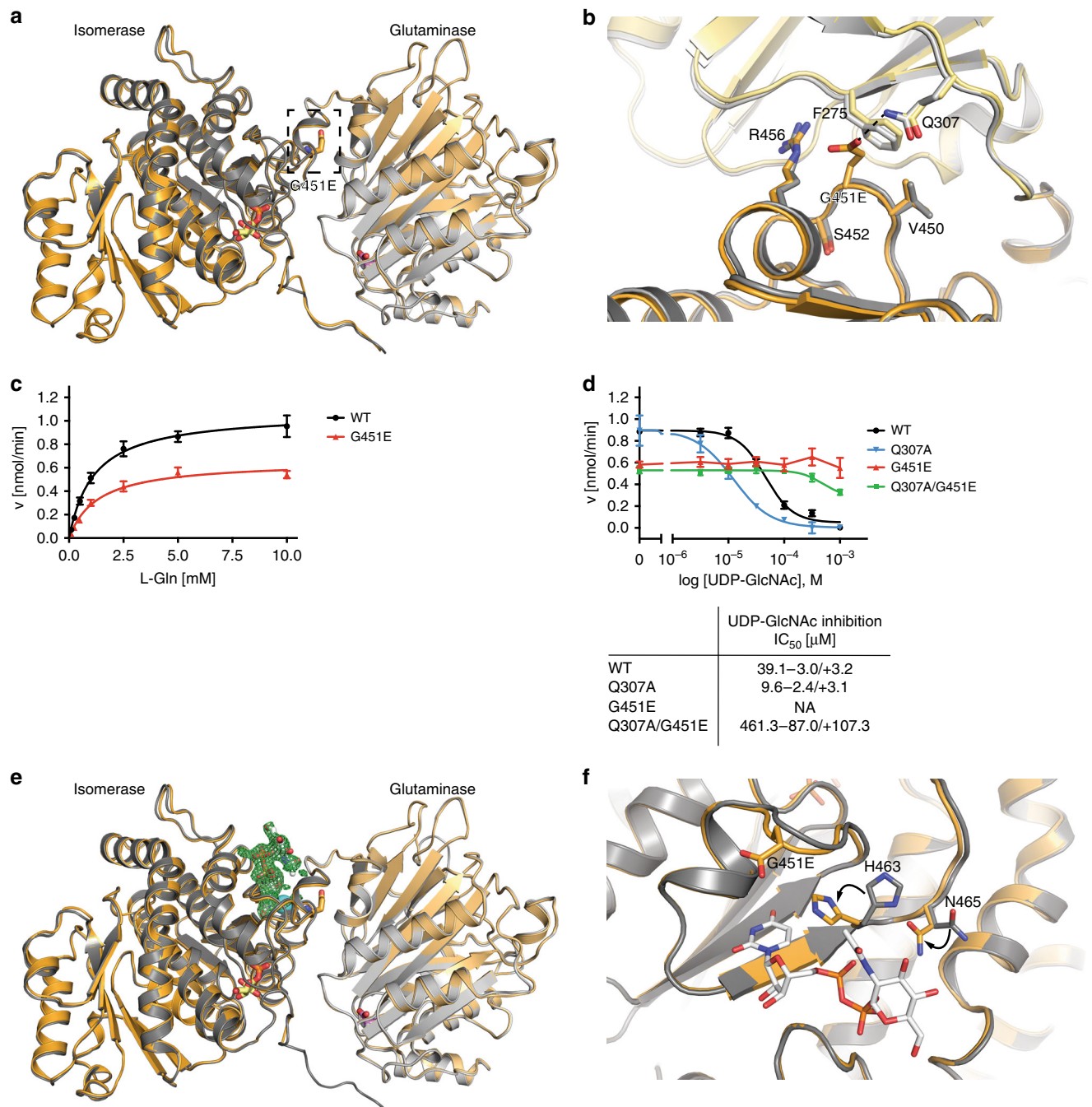

**Fig. 5 GFAT-1 gain-of-function mutation perturbs UDP-GlcNAc inhibition. a, b** Effect of G451E mutation on GFAT-1 structure. Proteins are presented as cartoons. Superposition of wild type GFAT-1 (light gray/dark gray) and G451E GFAT-1 (yellow/orange). Glc6P (yellow sticks) and L-Glu (violet sticks) are highlighted. **a** G451E (sticks) is located at the isomerase domain of GFAT-1 pointing towards the interdomain linker (black box). **b** Superposition of the wild type and the G451E GFAT-1 structure focusing on residues in close proximity to the mutation. **c** L-Gln kinetic of wild type (WT, black circle) and G451E (red triangle) GFAT-1 (mean ± SEM, $n = 5$). **d** Representative UDP-GlcNAc inhibition of wild type (black circle), G451E (red triangle), Q307A (blue triangle), and Q307A/G451E (green square) GFAT-1 (mean ± SD, $n = 3$). Table: IC$_{50}$ UDP-GlcNAc values (mean ± SEM, $n = 3$). **e, f** Superposition of UDP-GlcNAc-bound G451E GFAT-1 (yellow/orange) and wild type GFAT-1 in the absence of UDP-GlcNAc (gray). Proteins are presented as cartoons. Glc6P (yellow sticks), L-Glu (violet sticks), UDP-GlcNAc (white sticks), Mg$^{2+}$ (cyan sphere), and G451E (sticks) are highlighted. **e** Overall structure with Fo−Fc omit map (green) of UDP-GlcNAc binding to G451E GFAT-1 at a contour level of 3.0 RMSD. **f** Close-up of the UDP-GlcNAc-binding pocket with local side chain movements in G451E GFAT-1. Source data are provided as a Source Data file.

domain, leading to local structural changes. However, we did not observe any conformational changes at the glutaminase site, which would explain the inhibition of amidohydrolysing activity. Interestingly, while UDP-GlcNAc was a potent GFAT-1 inhibitor, its closely related epimer UDP-GalNAc emerged as a weak

GFAT-1 inhibitor. The epimers differ only in the orientation of the sugar's C4 hydroxyl group, suggesting that GFAT-1 differentiates between the epimers with high fidelity. The sugar points towards the interdomain linker, demonstrating the linker's key role in modulation of GFAT-1 activity. Characterization of a

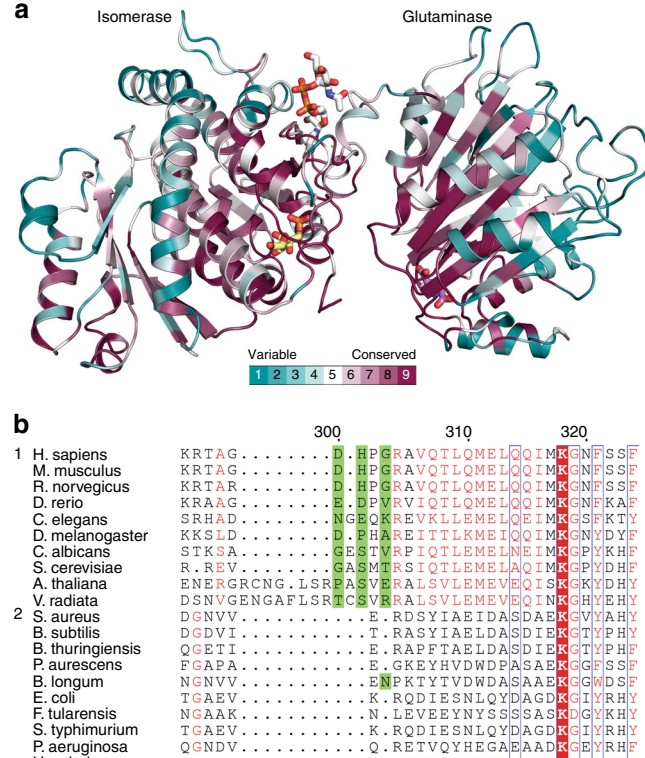

**Fig. 6 Conservation of GFAT-1. a** Overview of conservation in GFAT-1 monomer A. The protein is presented as a cartoon and colored according to the ConSurf color scheme from variable to conserved residues (teal to bordeaux). Glc6P (yellow sticks), L-Glu (violet sticks), and UDP-GlcNAc (white sticks) are highlighted. **b** Protein sequence alignment of the interdomain linker of GFAT-1. Group 1 represents eukaryotes and group 2 prokaryotes. Red boxes indicate identical residues; red letters indicate similar residues. Source data are provided as a Source Data file.

gain-of-function mutation, G451E, which causes HP activation and longevity in *C. elegans*, revealed that gain-of-function results from a drastically reduced sensitivity to UDP-GlcNAc inhibition. Finally, we showed that the gain-of-function mechanism leads to elevated HP flux in mammalian cells.

While the isomerase domain of the full-length structure is very similar to previously crystallized bacterial and eukaryotic isomerase domains[28,31,32], our structure of the eukaryotic glutaminase domain revealed major differences in extended β-sheets and longer loops connecting α-helices and β-sheets. These differences might allow for the design of antibacterial drugs targeting the glutaminase domain. Two phosphorylation sites, S235[22] and S243[24,25], are positioned within these extended loops, indicating their potential importance for the regulation of eukaryotic GFAT.

We observed an asymmetric GFAT-1 dimer, which allowed the characterization of an open and closed conformation of the isomerase and glutaminase domain relative to each other. In the open conformation observed in monomer A, L-Glu was bound to GFAT-1, which stabilizes an inactive orientation of the catalytic Cys2 pointing away from the active site[27], as well as a Q-loop movement closing the glutaminase site and inducing a rotation of Trp96. Previously, the Q-loop movement was reported for the *E. coli* enzyme, where after binding of the competitive inhibitor DON, an analog of the glutamine substrate, the rotation of Trp96 equivalent Trp74 opens the ammonia channel[17]. In contrast, no ammonia channel was formed after L-Glu binding in our structure. These data suggest that L-Glu binding induces some changes similar to L-Gln binding, but as it is the product and not substrate

of the reaction, it switches Cys2 to a catalytic incompetent rotamer and does not provoke the formation of the ammonia channel. Although L-Glu was present in high excess, it was not detected in monomer B, where the Q-loop does not seal the glutaminase active site. Here, the L-Glu product has probably diffused out of the active site owing to its greater accessibility to the solvent space.

Moreover, we describe a glutamate-free GFAT-1 structure where the R-loop at the glutaminase site of monomer A showed major structural changes, flipping into the cleft between glutaminase and isomerase site. The conserved R-loop arginines were previously suggested to be key elements in maintaining a functional glutaminase site and to be involved in interdomain communication in bacterial GFAT and other Ntn-hydrolases (PURF, ASNB)[17,27,44,45]. In *E. coli* residue Arg26, which is the equivalent of Arg33 in the human enzyme, is thought to keep Cys2 in an active position pointing towards the active site after glutamine binding and was suggested to mediate communication between the two active sites[17]. Our study reveals structural evidence for a direct interaction of Arg33 from the glutaminase site with Asp667 of the isomerase domain, which is in close proximity to the catalytically important C-loop (residues 670–681). Presumably, the salt bridge between Arg33 and Asp667 keeps the two domains in an open conformation. After disruption of the salt bridge by reorientation of the R-loop upon glutamine binding, the cleft could close. This allows the C-loop to interact with both active sites, which is necessary for catalysis. Thereby, the R-loop might signal the presence of substrate in the glutaminase active site to the isomerase domain. Taken together, these results clearly point to an important role of Arg33 in interdomain communication during the GFAT-1 catalytic cycle.

To understand GFAT-1 regulation, UDP-GlcNAc and UDP-GalNAc were soaked into GFAT-1 crystals. We detected binding to both monomers and observed local structural changes at the binding site of the isomerase domains. However, binding had no consequences at the glutaminase site. The two active sites of GFAT-1 are coupled and GFAT-1 function depends on the communication and relative orientation of the two domains. For the bacterial homolog GlmS, it is reported that the glutaminase domain adopts a fixed position relative to the isomerase domain upon Frc6P binding, and that the glutaminase function is activated in the presence of Frc6P by 100-fold[27,41]. The Frc6P-dependent activation of the glutaminase activity was also reported for the *C. albicans* homolog[34]. It is therefore very likely that this also happens in the human enzyme. We propose that UDP-GlcNAc could disturb the tight coupling of the active sites by interference with the orientation of the two domains relative to each other. There are several lines of evidence that support a key role of the interdomain linker in this process. First, activity assays comparing UDP-GlcNAc and UDP-GalNAc inhibition revealed that the orientation of the sugar's C4 hydroxyl group, which is positioned to interact with the interdomain linker, is sufficient to modulate the inhibition (Fig. 7g). Second, in the G451E gain-of-function mutant, which showed a reduced sensitivity to UDP-GlcNAc-dependent feedback inhibition, Glu451 interacts with Gln307 from the interdomain linker and the Q307A mutant showed an increased sensitivity to UDP-GlcNAc. Third, while both active sites are highly conserved from prokaryotes to eukaryotes, the interdomain linker is only conserved among eukaryotes, whose GFAT-1 is susceptible to UDP-GlcNAc inhibition. Fourth, previous publications indicated a specific role of the N-acetyl moiety of UDP-GlcNAc in inhibition, which points towards the interdomain linker: Assir et al.[33] reported that UDP and UDP-Glc do not inhibit GFAT-1, but are able to bind with a similar $K_D$ as for UDP-GlcNAc. Moreover, Walter et al.[46] generated metabolic chemical reporters with large azide- or alkyne-

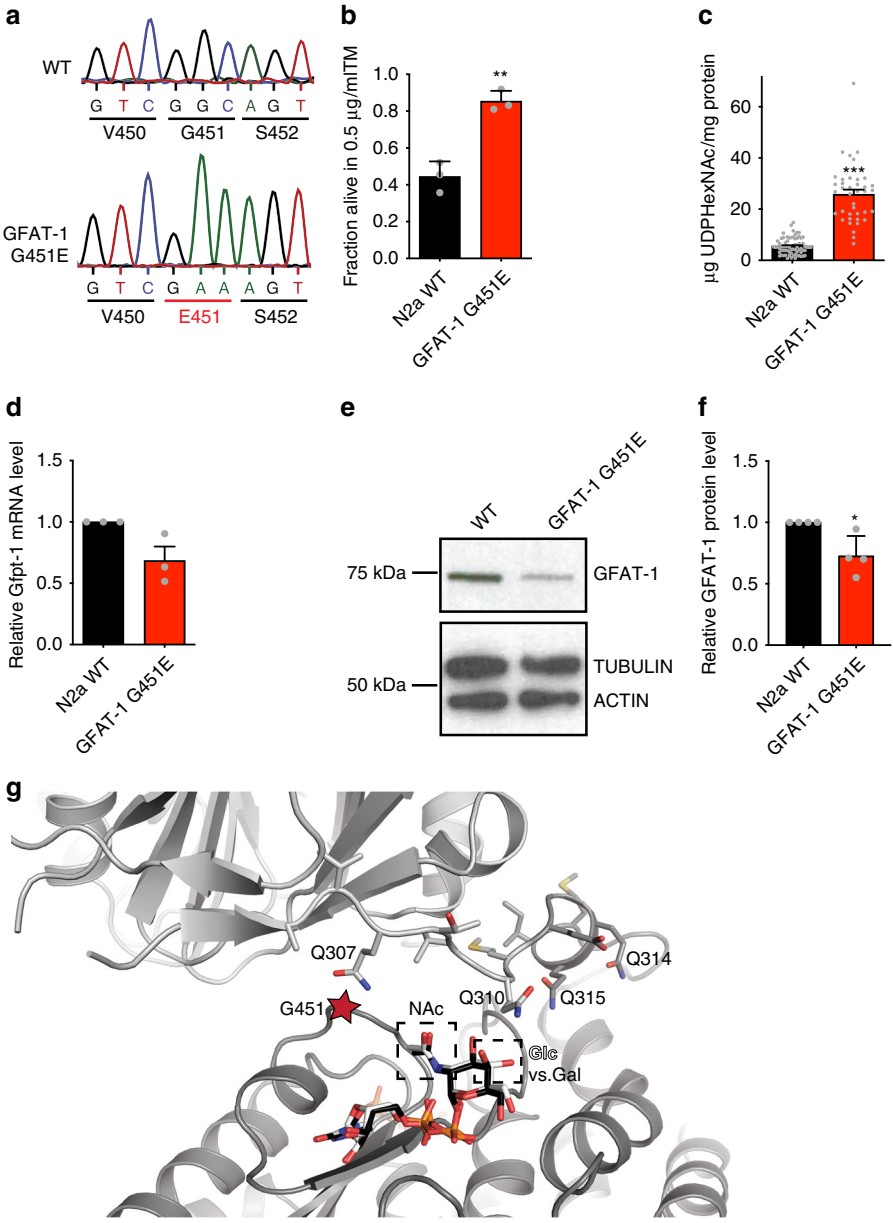

**Fig. 7 The G451E substitution activates GFAT-1 in mammalian cells. a** Sanger sequencing results of wild type (WT) and G451E genomic *Gfpt1* engineered in mouse neuroblastoma cells (N2a). **b** Cell viability (XTT assay) of WT and GFAT-1 G451E-engineered N2a cells after a 48 h treatment with 0.5 μg/ml tunicamycin (TM) (mean + SEM, $n = 3$, **$p < 0.01$, unpaired *t*-test). **c** UDP-HexNAc level in WT and GFAT-1 G451E N2a cells (mean + SEM, $n \geq 25$, ***$p < 0.001$, unpaired *t*-test). **d** Relative *Gfpt1* mRNA-level (qPCR) of GFAT-1 G451E N2a cells normalized to WT controls (mean + SEM, $n = 3$). **e, f** Western blot analysis of endogenous GFAT-1 protein levels from WT and GFAT-1 G451E N2a cells (**e**), including quantification relative to tubulin and the WT control cells (**f**, mean + SD, $n = 4$, *$p < 0.05$, unpaired *t*-test). **g** Close-up of the interdomain linker and UDP-GlcNAc-binding site in GFAT-1 monomer A. The protein is presented as a cartoon. The isomerase domain is colored in dark gray and the glutaminase domain is colored in light gray. UDP-GlcNAc (white sticks) and UDP-GalNAc (black sticks) are superimposed. Residues of the interdomain linker (sticks) and the position of G451 (red star) are highlighted. The NAc and C4 hydroxyl groups of UDP-GlcNAc/UDP-GalNAc are pointing towards the interdomain linker (black boxes). Source data are provided as a Source Data file.

residues at the N-acetyl position, which also failed to inhibit GFAT-1[46]. Together, these publications emphasize the role of the N-acetyl group in inhibition whose only possible interaction partner are residues from the interdomain linker, especially Gln307 (Fig. 7g). Taken together, our current study suggests a previously unknown role of the interdomain linker in UDP-GlcNAc inhibition.

While the orientation of the two domains with their respective active sites is relevant for catalysis, the correct orientation of the UDP-GlcNAc-bound isomerase and the interdomain linker seem

to be important for UDP-GlcNAc inhibition. In our mutants, we observed altered sensitivities to UDP-GlcNAc, which we interpret as follows: in the gain-of-function mutant G451E, which showed a reduced sensitivity to UDP-GlcNAc inhibition, the hydrogen bond between Glu451 and Gln307 stabilizes the linker and both residues might function as spacers between the interdomain linker and the UDP-GlcNAc-bound isomerase domain, preventing a close interaction. Furthermore, the G451E mutation might interfere with UDP-GlcNAc binding as inhibition can only be observed at very high concentrations. The enhanced sensitivity to

UDP-GlcNAc of mutant Q307A might be due an increased flexibility of the interdomain linker and the smaller side chain would allow a closer interaction of the interdomain linker with the UDP-GlcNAc-bound isomerase domain. Interestingly, the double-mutant Q307A/G451E showed a ten times weaker response to UDP-GlcNAc than the wild type, thus indicating that the G451E substitution might be sufficient to disturb inhibition even in the presence of mutation Q307A. In the double mutant, the Glu451 side chain might disturb UDP-GlcNAc binding and prevent a close interaction between the UDP-GlcNAc-bound isomerase domain with the interdomain linker.

In mammalian cells, loss of regulation by UDP-GlcNAc feedback inhibition upon G451E mutation constitutively activated GFAT-1, which increased HP flux. Similar to the situation in the nematode *C. elegans*, this increased HP flux goes along with elevated tolerance to tunicamycin-induced ER stress. Notably, GFAT-1 protein and mRNA levels were reduced in the mutant compared to wild type control cells, demonstrating that the observed increase in HP flux stems from higher GFAT-1 activity, not abundance. Since GFAT-1 G451E activity in vitro is not increased compared to wild type GFAT-1, our cellular data indicate that wild type GFAT-1 is nearly completely inhibited by UDP-GlcNAc under physiological conditions. This constant inhibition in vivo positions loss of feedback inhibition as the physiologically relevant activation mechanism of the HP. In light of previous data from *C. elegans*, where elevated HP flux counters toxic protein aggregation and extends lifespan[4], activation of GFAT-1 by interfering with feedback inhibition constitutes a promising target to ameliorate proteotoxic diseases. This work is an important contribution to the structural and functional characterization of GFAT-1, which might inform the development of GFAT-1 modulating molecules.

## Methods

**Plasmids and site-directed mutagenesis**. An internal His₆-tag was introduced into human GFAT-1 isoform 2[47] between Gly299 and Asp300 in a FLAG-HA-hGFAT-1-pcDNA3.1 plasmid (pcDNA™3.1$^{(+)}$; Thermo Fisher Scientific #V79020) by site-directed mutagenesis[48] using the following primers: hGFAT1-299-His6_for cgaactgcaggaCATCACCATCACCATCACgatcaccccggcgagctgtgcaaaactccagatggaac and hGFAT1-299-His6_rev cgtccggggtgatcGTGATGGTGATGGTGATGtcctgcagttcgt ttaattcgatggatagaaagacgtcca.

The hGFAT-1 gene with internal His₆-tag was subsequently subcloned into the pFL vector using XbaI and HindIII entry sites. The mutations Q307A, G451E, and G461E were introduced into pFL-hGFAT1-His299 by site-directed mutagenesis as described previously[48] (primers: hGFAT1_Q307A_for GAGCTGTGGgctACACTCC AGATGGAACTCC, hGFAT1_Q307A_rev GGAGTGTagcCACAGCTCGTCCGG GG, hGFAT1_G451E_for CACAGTTGaaAGTTCCATATCACGGGAG, hGFA T1_G451E_rev GGAACTttCAACTGTGTTTGTGATCCCC, hGFAT1_G461E_for CAGATTGTGaAGTTCATATTAATGCTGGTC, hGFAT1_G461E_rev GAACTtC ACAATCGTGTCTCCCGTGATATG).

**Baculovirus generation and insect cell expression**. Sf21 (DSMZ no. ACC 119) suspension cultures were maintained in SFM4Insect™ HyClone™ medium with glutamine (GE Lifesciences) in shaker flasks at 27 °C and 90 r.p.m. in an orbital shaker. GFAT-1 variants were expressed in Sf21 cells using the MultiBac baculovirus expression system[37]. In brief, hGFAT-1 variants (from the pFL vector) were integrated into the baculovirus genome via Tn7 transposition and maintained as bacterial artificial chromosome in DH10EMBacY *E. coli* cells. Recombinant baculoviruses were generated by transfection of Sf21 with bacmid DNA. The obtained baculoviruses were used to induce protein expression in Sf21 cells.

**GFAT-1 purification**. Sf21 cells were lysed by sonication in lysis buffer (50 mM Tris/HCl pH 7.5, 200 mM NaCl, 10 mM Imidazole, 2 mM tris(2-carboxyethyl) phosphin (TCEP), 0.5 mM Na₂Frc6P, 10% (v/v) glycerol, supplemented with complete EDTA-free protease inhibitor cocktail (Roche) and 10 µg/ml DNAseI (Sigma)). Cell debris and protein aggregates were removed by centrifugation and the supernatant was loaded on an Ni-NTA Superflow affinity resin (Qiagen). The resin was washed with lysis buffer and the protein eluted with lysis buffer containing 200 mM Imidazole. Subsequently, proteins were separated according to their size on a HiLoad™ 16/60 Superdex™ 200 prep grade prepacked column (GE Healthcare) using an ÄKTAprime chromatography system at 4 °C with a SEC buffer containing 50 mM Tris/HCl, pH 7.5, 2 mM TCEP, 0.5 mM Na₂Frc6P, and

10% (v/v) glycerol. For co-crystallization of GFAT-1 with GlcN6P, GFAT-1 was purified in the absence of 0.5 mM Na₂Frc6P.

**Crystallization and crystal soaking**. For crystallization experiments, the SEC buffer was supplemented with 50 mM L-Arg and 50 mM L-Glu to improve protein solubility[49]. GFAT-1 was crystallized at a concentration of 8 mg/ml in sitting-drops by vapor diffusion at 20 °C. Initial crystals grew in the PACT *premier*™ HT-96 (Molecular Dimensions) screen with a reservoir solution containing 0.1 M Bis tris propane pH 8.5, 0.2 M potassium sodium tartrate and 20% (w/v) PEG3350 and were further optimized. The optimization screen was set up with drop ratios of 1.5 µl protein solution to 1.5 µl precipitant solution and 2 µl protein solution to 1 µl precipitant solution. Best crystals grew in a broad range of 0.1 M Bis tris propane pH 8.5–9.0, 0.2–0.4 M potassium sodium tartrate, and 20% (w/v) PEG3350. For UDP-GlcNAc/UDP-GalNAc soaking experiments, crystals were soaked with 5 mM UDP-GlcNAc or UDP-GalNAc in reservoir solution for 2 h. Glu-dilution was performed in 40-steps by exchanging 25% of the mother liquor with reservoir solution without L-Arg and L-Glu supplemented with 1 mM Frc6P. Co-crystallization with GlcN6P was achieved by addition of 1 mM GlcN6P to the protein solution. Data were collected from crystals cryoprotected with reservoir solution supplemented with 15% (v/v) glycerol.

**Data collection and refinement**. X-ray diffraction measurements were performed at beamline P13 at PETRA III, DESY, Hamburg (Germany), beamline X06DA at the Swiss Light Source, Paul Scherrer Institute, Villigen (Switzerland), or beamline ID30A-3 at the European Synchrotron Radiation Facility (ESRF), Grenoble (France). The human full-length GFAT-1 structure was determined by molecular replacement with phenix.phaser[50,51] using the model of the human GFAT-1 isomerase domain (PDB 2ZJ3) as search model. After a first round of autobuilding using the ARP/wARP Web Service[52] GFAT-1 was further manually built using COOT[53] and iterative refinement rounds were performed using phenix.refine[51]. One of the glutaminase domains (chain B) was not well defined in the structure. After placing initial strands, the domain was completed by superposition with the glutaminase domain of chain A. Structures of GFAT-1 variants and UDP-GlcNAc/UDP-GalNAc soaked crystals were solved by molecular replacement using the full-length GFAT-1 structure as a search model. Geometry restraints for ligands were generated with phenix.elbow software[51] or the Grade Web Server. Structures were visualized using PyMOL (Schrödinger) and 2D ligand–protein interaction diagrams were generated using LigPlot+[54].

**Alignments**. Protein sequence alignments were created with Clustal Omega[55], the ESPript3 server[56], and further modified. Most organisms for the alignment were chosen according to publications about GFAT, which were found in the BRENDA enzyme database (UnitProt IDs: *Homo sapiens* isoform 2: Q06210-2, *Mus musculus* isoform 2: P47856-2, *Rattus norvegicus*: P82808; *Caenorhabditis elegans*: Q95QM8, *Drosophila melanogaster*: Q7PLC7, *Candida albicans*: P53704, *Saccharomyces cerevisiae*: P14742, *Arabidopsis thaliana*: Q9LIP9, *Staphylococcus aureus*: X5E1D5, *Bacillus subtilis*: P0CI73, *Bacillus thuringiensis*: A0A0B5NV56, *Paenarthrobacter aurescens*: A1R8P7, *Bifidobacterium longum*: Q8G545, *Escherichia coli*: P17169, *Francisella tularensis*: A0A0E3A6K1, *Salmonella typhimurium*: Q8ZKX1, *Pseudomonas aeruginosa*: Q9HT25, *Helicobacter pylori*: O26060; NCBI IDs: *Danio rerio*: NP_001029093.1, *Vigna radiata* var. *radiata*: XP_014523177.1). The ConSurf server[43] was used to highlight conserved regions in the structure.

**Protein mass spectrometry (LC-MS/MS)**. Peptide mass fingerprinting was performed to analyze the amino acid sequence, and phosphorylation of purified GFAT-1. For this, 5 µg GFAT-1 was alkylated by chloroacetamide, reduced with TCEP, and digested by Trypsin (Promega, MS grade) over night at 37 °C. The resulting peptides were purified using C-18 StageTips.

Peptides were separated on a 25 cm, 75 µm internal diameter PicoFrit analytical column (New Objective) packed with 1.9 µm ReproSil-Pur 120 C18-AQ media (Dr. Maisch HPLC GmbH) using an EASY-nLC 1200 (Thermo Fisher Scientific). The column was operated at 50 °C. Buffer A and B were 0.1% formic acid in water and 0.1% formic acid in 80% acetonitrile. Peptides were separated on a segmented gradient from 6% to 31% buffer B for 30 min and from 31% to 50% buffer B for 5 min at 200 nl/min. Eluting peptides were analyzed on a QExactive HF mass spectrometer (Thermo Fisher Scientific). Peptide precursor *m/z* was measured at 60000 resolution in the 300–1500 *m/z* range. The top eight most intense precursors with charge state from 2 to 6 only were selected for HCD fragmentation using 27% normalized collision energy. The *m/z* values of the peptide fragments were measured at a resolution of 30,000 using a minimum AGC target of 1e6 and 100 ms maximum injection time. Upon fragmentation, precursors were put on a dynamic exclusion list for 40 s.

The raw data were analyzed with MaxQuant version 1.5.2.8 using the integrated Andromeda search engine[57,58]. Peptide fragmentation spectra were searched against manually created GFAT-1 fasta sequence database. The database was automatically complemented with sequences of contaminating proteins by MaxQuant. Methionine oxidation and protein N-terminal acetylation were set as variable modifications; cysteine carbamidomethylation was set as fixed modification. The digestion parameters were set to "specific" and "Trypsin/P." The

minimum number of peptides and razor peptides for protein identification was 1; the minimum number of unique peptides was 0. Protein identification was performed at a peptide spectrum matches and protein false discovery rate of 0.01. The "second peptide" option was on. Extracted ion chromatograms were generated using Qual Browser version 2.2. Data visualization was done using ggplot2.

**GDH-coupled activity assay and UDP-GlcNAc inhibition**. GFAT's amidohydrolysis activity was measured with a coupled enzymatic assay using bovine glutamate dehydrogenase (GDH, Sigma Aldrich G2626) in 96-well standard microplates (F-bottom, BRAND #781602) as previously described[36] with small modifications. In brief, the reaction mixtures contained 6 mM Frc6P, 1 mM APAD, 1 mM EDTA, 50 mM KCl, 100 mM potassium-phosphate buffer pH 7.5, 6.5 U GDH per 96-well and for L-Gln kinetics varying concentrations of L-Gln. For UDP-GlcNAc/UDP-GalNAc inhibition assays the L-Gln concentration was kept at 10 mM. The plate was pre-warmed at 37 °C for 10 min and the activity after enzyme addition was monitored continuously at 363 nm in a microplate reader. The amount of formed APADH was calculated with $\varepsilon_{(363\,nm,APADH)} = 9100\,l\,mol^{-1}\,cm^{-1}$. Reaction rates were determined by Excel (Microsoft) and $K_m$, $v_{max}$, and $IC_{50}$ were obtained from Michaelis Menten or dose response curves, which were fitted by Prism 7 or 8 software (Graphpad).

**GNA-1 cloning**. In order to remove a second *Nde*I restriction site, a silent mutation, H77H, was introduced into human GNA-1 in the FLAG-HA-hGNA1-pcDNA3.1 plasmid by site-directed mutagenesis[48] (primers: hGNA1-H77H_for cttttgagcaCatgaagaaatctgggg, hGNA1-H77H_rev cttcatGtgctcaaaagatttcataaattgttc). Subsequently, GNA-1 was cloned into the pET28a expression vector (Merck Millipore) using *Nde*I and *Hin*dIII restriction sites (primers: hGNA1_NdeI_FOR gagCATATGatgaaacctgatgaaactcctatgtttgaccc, hGNA1_HindIII_REV gagAAGCTTtcactttagaaacctccgacacatgtag).

**GNA-1 expression and purification**. Human GNA-1 with N-terminal His$_6$-tag was expressed in Rosetta (DE3) *E. coli* cells. LB cultures were incubated at 37 °C and 180 r.p.m. until an OD$_{600}$ of 0.4–0.6 was reached. Then, protein expression was induced by addition of 0.5 mM isopropyl-β-D-1-thiogalactopyranosid and incubated for 3 h at 37 °C and 180 r.p.m. Cultures were harvested and pellets stored at −80 °C. Human GNA-1 purification protocol was adopted from Hurtado-Guerrero et al.[59] with small modifications. *E. coli* were lysed in 50 mM HEPES/NaOH pH 7.2, 500 mM NaCl, 10 mM Imidazole, 2 mM 2-mercaptoethanol, 5% (v/v) glycerol with complete EDTA-free protease inhibitor cocktail (Roche), and 10 μg/ml DNAseI (Sigma) by sonication. The lysate was clarified by centrifugation and the supernatant loaded on Ni-NTA Superflow affinity resin (Qiagen). The resin was washed with wash buffer (50 mM HEPES/NaOH pH 7.2, 500 mM NaCl, 50 mM Imidazole, and 5% (v/v) glycerol), and the protein was eluted with wash buffer containing 250 mM imidazole. Eluted protein was then dialyzed against storage buffer (20 mM HEPES/NaOH pH 7.2, 500 mM NaCl, 5% (v/v) glycerol).

**GNA-1 and GNA-1-coupled activity assays**. The activity of human GNA-1 was measured in 96-well standard microplates (F-bottom, BRAND #781602) as described previously[60]. For kinetic measurements, the assay mixture contained 0.5 mM AcCoA, 0.5 mM DTNB, 1 mM EDTA, 50 mM Tris/HCl pH 7.5, and varying concentrations of D-GlcN6P. The plates were pre-warmed at 37 °C and reactions were initiated by addition of GNA-1. The absorbance at 412 nm was followed continuously at 37 °C in a microplate reader. The amount of produced TNB, which matches CoA production, was calculated with $\varepsilon_{(412\,nm,TNB)} = 13800\,l\,mol^{-1}\,cm^{-1}$. Typically, GNA-1 preparations showed a $K_m$ of 0.2 ± 0.1 mM and a $k_{cat}$ of 41 ± 8 s$^{-1}$.

GFAT's D-GlcN6P production was measured in a GNA-1-coupled activity assay following the consumption of AcCoA at 230 nm in UV transparent 96-well microplates (F-bottom, Brand #781614) as described by Li et al.[60]. In brief, the assay mixture contained 10 mM L-Gln, 0.1 mM AcCoA, 50 mM Tris/HCl pH 7.5, 2 μg hGNA-1, and varying concentrations of Frc6P. The plates were incubated at 37 °C for 4 min and reactions started by adding L-Gln. Activity was monitored continuously at 230 nm and 37 °C in a microplate reader. The amount of consumed AcCoA was calculated with $\varepsilon_{(230\,nm,AcCoA)} = 6436\,l\,mol^{-1}\,cm^{-1}$. As UDP-GlcNAc absorbs light at 230 nm, the GNA-1-coupled assay cannot be used to analyze UDP-GlcNAc effects on activity.

**Mammalian cell maintenance and viability assays**. N2a mouse neuroblastoma cells (ATCC) were cultured in DMEM containing 1 g/l glucose (Gibco) supplemented with 10% fetal bovine serum (Life technologies). Relative cell viability was assessed using the XTT cell proliferation Kit II (Roche) according to the manufacturer's instructions. Treatment with 0.5 μg/ml tunicamycin was performed for 48 h, starting 24 h after cell seeding. XTT turnover was normalized to untreated control cells.

**Gene editing and genotyping by Sanger sequencing**. The specific GFAT-1 G451E substitution was engineered in N2a cells using the CRISPR/Cas9 technology as described previously[61]. DNA template sequences for small-guide RNAs were designed online (http://crispor.org), purchased from Sigma, and cloned into the Cas9-GFP-expressing plasmid PX458 (addgene #48138, Guide1: GAGTCGG-CAGTTCTATATCA, Guide 2: GGTGGGGATCACAAATACAGT). Corresponding guide and Cas9-expressing plasmids were co-transfected with a single-stranded DNA repair template (Integrated DNA Technologies, GGCGAGACAGCTGACA CCCTGATGGGGACTTCGTTACTGTAAGGAGAGAGGGGCCTTAACTGTGGG CATCACTAATACAGTCGAAAGTTCCATATCAAGAGAGACAGATTGCGGG GTTCATATTAATGCTGGTCCTGAGATTGGCGTGGCCAGTACAAAG) using Lipofectamine 3000 (Thermo Fisher Scientific) according to the manufacturer's instructions. GFP-positive cells were singled using a FACS machine and subjected to genotyping. DNA was extracted (DNA extraction solution, Epicentre Biotechnologies) and edited regions were specifically amplified by PCR (for primer AGTCGGTTGGTTTTTCGTGT, rev primer ACTGCCCCACAGATCAGAGT). Sanger sequencing was performed at Eurofins Genomics GmbH.

**RNA isolation and qRT-PCR**. Cells were collected in QIAzol (Qiagen) and frozen in liquid nitrogen. Total RNA was isolated using the RNeasy Mini Kit (Qiagen) and cDNA was subsequently generated by iScript cDNA Synthesis Kit (BioRad). qRT-PCR was performed with Power SYBR Green master mix (Applied Biosystems) on a ViiA 7 Real-Time PCR System (Applied Biosystems). GAPDH expression functioned as an internal control. GFAT primer (5′→3′): for AAAGGAAGCTGC GGTCTTTCCC, rev GTGTGCTCTATCACGGCACTTG; GAPDH primer: for GGCATGGACTGTGGTCATGAG; rev TGCACCACCAACTGCTTAGC.

**Antibodies**. The following antibodies were used in this study: GFAT1 (rb, EPR4854, Abcam ab125069, 1:1000), α-tubulin (ms, DM1A, Sigma T6199, 1:50000), β-ACTIN (ms, 8H10D10, Cell Signaling 3700S, 1:50000), rabbit IgG (gt, Life Technologies G21234, 1:5000), and mouse IgG (gt, Life Technologies G21040, 1:5000).

**Small-molecule LC/MS/MS analysis**. UDP-HexNAc concentrations were measured as described previously[4]. In brief, cells were trypsinized, lysed in water by freeze/thaw cycles, and subjected to chloroform/methanol extraction. Absolute UDP-HexNAc levels were determined using an Acquity UPLC connected to a Xevo TQ Mass Spectrometer (both Waters) and normalized to total protein content.

**Reporting summary**. Further information on research design is available in the Nature Research Reporting Summary linked to this article.

## Data availability
Structural data reported in this study have been deposited in the Protein Data Bank with the accession codes 6R4E, 6R4F, 6R4G, 6R4H, 6R4I, 6R4J, 6SVM, 6SVO, 6SVP, 6SVQ. All other data supporting the presented findings are available from the corresponding authors upon reasonable request. The source data underlying Figs. 4f–g, 5c, d, 6b, 7b–f, and Supplementary Figs. 4g, 4i–j, 5b, c, 5e, f, 6b–d are provided as a Source Data file.

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

## Acknowledgements

We thank all M.S.D. and U.B. laboratory members for helpful discussions. The FLAG-HA-hGFAT-1-pcDNA3.1 and FLAG-HA-hGNA1-pcDNA3.1 plasmids were kindly provided by Christoph Geisen (Max Planck Institute for Biology of Ageing). We are grateful to Schirin Birkmann for support in the insect cell maintenance. Crystals were grown in the Cologne Crystallization facility ($C_2f$). We thank the staff of beamline P13 at PETRA III, DESY, Hamburg (Germany), beamline X06DA at the Swiss Light Source, Paul Scherrer Institute, Villigen (Switzerland) and beamline ID30A-3 at the European Synchrotron Radiation Facility (ESRF), Grenoble (France) for their support during data collection. Funding for some synchrotron visits was provided by the iNEXT initiative (EU program Horizon 2020). We thank I. Atanassov and X. Li from the proteomics core facility of the Max Planck Institute for Biology of Ageing. We thank K. Folz-Donahue, L. Schumacher, A. Just, and C. Kukat from the FACS and imaging core facility of the Max Planck Institute for Biology of Ageing. We thank Y. Hinze and P. Giavalisco from the metabolomics core facility of the Max Planck Institute for Biology of Ageing. This work was supported by the German Federal Ministry of Education and Research (BMBF, grant 01GQ1423A EndoProtect), by the German Research Foundation (DFG, Projektnummer 73111208-SFB 829, B11 and B14), and by the European Commission (ERC-2014-StG-640254-MetAGEn). The Cologne Crystallization Facility $C_2f$ was supported by DFG grant INST 216/949-1 FUGG.

## Author contributions

S.R., U.B. and M.S.D. designed the project. S.R. performed all biochemical and crystallization experiments. M.H. performed all experiments related to CRISPR/Cas9 and mammalian cell culture. C.P. supported the biochemical and crystallization experiments. K.A. helped with the tissue culture experiments. S.R., M.H., M.S.D. and U.B. wrote the manuscript. S.R. prepared the figures.

## Competing interests

The authors declare no competing interests.
