## [Peer Review File · Nature Communications]

Reviewers' comments:

Reviewer #1 (Remarks to the Author):

Ruegenberger et al. report here the first full-length crystal structure at 2.4 Å resolution of human GFAT-11, an important enzyme of the hexosamine pathway. While co-crystallization with the allosteric inhibitor UDP-GlcNAc, which inhibits the glutaminase function, did not lead crystals, UDP-GlcNAc was successfully soaked into the GFAT-1 crystals. However, no structural changes of the glutaminase active site upon binding UDP-GlcNAc were observed, which could have explained feedback inhibition, probably because a larger panel of crystallization conditions needs to be screened to trap a conformation showing how UDP-GlcNAc binding triggers conformational changes in the glutaminase site.

The structures of the G461E and G451E mutants are also reported, together with their kinetic constants and IC50 constants for inhibition by UDP-GlcNAc. Both glycines 461 and 451 are located at the bottom of the UDP-GlcNAc binding pocket. The mutants had kinetic constants similar to those of the wild-type protein and did not respond to UDP-GlcNAc feedback inhibition. While UDP-GlcNAc could not be soaked into the G461E crystals, it could bind to the G451E crystals. The G451E mutation has been shown previously to lead to gain-of-function in *C. elegans* (Denzel et al. Cell, 2014) i. e. it leads to elevated hexosamine flux and extended lifespan. Moreover, N2a mouse neuroblastoma cell lines bearing the G451E mutation were shown to be more resistant to tunicamycin and to have an increased level of UDP-N-acetyl-hexosamine compared to cell lines bearing wild-type GFAT-1. As GFAT-1 expression was decreased upon the G451E mutation, the gain-of-function likely results from enhanced GFAT-1 activity.

From their results, the authors conclude that the gain-of-function in the G451E mutant might result from impaired transmission of the UDP-GlcNAc-mediated feedback inhibition from its binding pocket to the site of L-Gln hydrolysis and that the interdomain linker might be involved in signal transmission.

In conclusion, despite this paper reports the first crystal structure of full-length human GFAT-1, which is interesting to a broad scientific audience as activation of GFAT-1 appears to be a promising target to ameliorate proteotoxic diseases, and is technically sound, it does not answer questions about the feedback inhibition mechanism by UDP-GlcNAc, which is the main hallmark of the mammalian enzyme compared to the bacterial enzyme. Although in vivo experiments indicate that point mutations that alter inhibition by the allosteric inhibitor UDP-GlcNAc activate the hexosamine pathway, extending lifespan and limiting ageing-associated pathologies, the structural basis for this activation needs to be consolidated. Therefore I do not recommend publication of this article in the present state by Nature Communications.

Major modifications

A) Oligomerization

p.6 lines 21-28

"and is in accordance with the estimated molecular weight from the gel filtration step". The value should be given as well as the theoretical value for the dimer. However, given that GFAT-1 displays an elongated shape, calibration with standards of a SEC column is not enough to determine the molar mass and the oligomerization state of the protein in solution. So I recommend to delete this part of the sentence. The molar mass in solution is given by DLS. Moreover, I suggest to invert the results in solution and the crystallographic data, as follows: This dimeric assembly is judged as stable in solution by the PISA server. Yet, the molar mass measured by DLS (323 kDa) indicates a tetrameric structure in solution (monomer molar mass of 78.8 kDa) (Supplementary Fig. 1d). Accordingly, the crystal structure shows that two rather weakly interacting dimers could form a tetramer similar to the one described for the structure of the isolated isomerase domains of the *Candida albicans* ortholog (Supplementary Fig. 1b,c)30.

However, this tetramer appears to be less stable than the dimer because the PISA server did not reveal a stable tetramer from crystal packing.

B) Electron density maps

Globally, electron density maps omitting the important regions that are discussed are lacking. Moreover, as part of Nature journals publishing policies for macromolecular crystallographic data, a stereo image of a portion of the electron density map should be provided with the submitted manuscript. None is provided.

p. 7 line 25 "corresponding electron density..." and Figure 1d and 1e

An electron density map omitting Frc6P and Glu contoured at 2.5-3 sigma should be superposed onto the active sites. In particular, it is important to see that the electron density in the isomerase site fits a linear and not a cyclic sugar.

p. 7 lines 15-16 and Sup Figure 1F

"S243 was found phosphorylated in both mass spectrometry analysis and the crystal structure (Supplementary Fig. 1a,f)."

An electron density map omitting phosphorylated serine contoured at 2.5-3 sigma should be shown instead of the 2Fo-Fc map ((Supplementary Fig. 1f).

p. 8 lines 28-30

"The glutaminase domain of monomer B makes no crystal contacts and is more flexible resulting in partially low electron density. (Supplementary Fig. 2a,b)" No electron density is shown in Sup Figure 2a and 2b.

p. 11 lines 22-24

"Indeed, UDP-GlcNAc was not bound to the G461E mutant of GFAT-1 after crystal soaking (Fig. 4e)". A 2Fo-Fc or Fo-Fc map should be shown to illustrate this statement.

Supp Figure 3.

Moreover, Fo-Fc maps omitting Glu and the Q-loop (and not 2Fo-Fc maps) should be drawn.

Finally, the omit electron density maps should be shown both for the Glu-bound and the Glu-free structures for comparison.

Figure 5f. A Fo-Fc omit map should be shown.

Concerning the molecule bound at the isomerase site, there is an inconsistency between the text (Glc6P) and the figure legends (Frc6P):

p. 7 lines 30-31 : "we decided to model the product Glc6P in our crystal structures"

figure legends (eg p. 31 line 29: "Frc6P (yellow sticks)"

C) Catalytic states of the structures

Discussion p. 14 lines 4-5

"In contrast to bacterial GFAT, human GFAT-1 forms an asymmetric dimer." This sentence should be omitted. Indeed, it is discussed below (lines 29 and following) and I don't think that it is a peculiarity of the human enzyme compared to the bacterial enzyme. I think that it is solely a different crystal packing that has frozen two conformational states for the monomer in the human enzyme, and not in the bacterial enzyme.

"For the first time, we observed a specific interaction between the two GFAT-1 domains relevant for the orchestrated coupling of the catalytic cycle."

It is not clear to me what is this specific interaction... Explain or delete

1) p. 9 lines 23-29

"Although both active sites are occupied by ligands in monomer A, no ammonia channel connecting the glutaminase and the isomerase site is formed."

"Thus, the conformation of monomer A might be a snapshot of the intermediate state after binding

of both substrates and before ammonia channel formation.”

2) p. 14 lines 31-34

“In monomer A, L-Glu was bound to GFAT-1, which was associated with conformational changes that would not allow GlcN6P synthesis.”

3) and p. 15 lines 2-4:

“This is in contrast to reports from the E. coli enzyme, where a rotation of Trp96 equivalent Trp74 opens the ammonia channel that is mainly formed by the C-loop17. These data suggest that L-Glu binding induces changes similar to L-Gln binding, but inactivates Cys2 and does not provoke the formation of the ammonia channel. Hence, the catalytic cycle cannot be completed and GFAT-1 is fixed in this state”.

These statements are wrong. There is a confusion between Glutamine (the substrate) and Glutamate (the product). In the E. coli structure (reference 17), the competitive inhibitor DON, an analog of the glutamine substrate (and not the Glu product) occupies the glutaminase active site and the Frc6P substrate occupies the isomerase site. Thus, the DON/Frc6P E. coli structure mimics a catalytic intermediate state of the enzyme, in which the ammonia channel is open. In contrast, in the human structure, L-Glu is the product and does NOT induce changes similar to substrate (L-Gln) binding. According to me, GFAT1 with Frc6P or Glc6P bound at the isomerase site and the L-Glu product bound at the glutaminase site does not correspond to a catalytic state.

“This indicates that the two monomers might be sequentially active”, NO

“meaning that the second monomer can bind to L-Gln only when the other monomer has completed its catalytic cycle.” NO: “meaning that in the second monomer the L-Glu product has already left the active site.”

p. 10 lines 18-19

“Together, the glutamate-free structure of monomer A suggests a communication of the glutaminase site through Arg33 with the isomerase domain, observed for the first time.”

“observed for the first time” is wrong. It should be replaced by “as observed for E. coli GlmS”

Indeed, in Reference 17, it was already shown that in E. coli GlmS, Arg26 (equivalent to Arg33 in GFAT-1) together with the C-terminal residues of the isomerase domain are major actors of the communication between the two active sites.

“ The communication between the two active sites is mediated by residues forming part of the channel, Arg-26 and the C-terminal residues including Thr-606” (last lines of the article).

p. 10 lines 24-25

“Co-crystallization of GFAT-1 with UDP-GlcNAc did not yield any crystals.”

It should be specified if only crystallization has been tried around the standard condition (0.1 M Bis tris propane pH 8.5, 0.2 M Potassium sodium tartrate and 20 % (w/v) PEG3350) or if a whole range of different conditions using various crystallization kits have been spanned.

p. 10 lines 26-28 and p. 32 line 23 Legend Figure 4.

An overview figure showing the respective location of Glu6P and UDP-NAc-GlcN6P in the isomerase domain is lacking. Also, it is not clear if Glc6P or Frc6P is present in UDP-NAc-GlcN6P-bound GFAT-1: Frc6P in yellow sticks is not seen in Figure 4. If present, change the orientation or show a stereo figure!

p. 13 line 5-6

“Furthermore, the interdomain linker might be involved in signal transmission.” I think this statement is an over-interpretation of the crystallographic data.

Minor modifications

Abstract line 1 and p. 4 line 2

“amidotransferase” and not “aminotransferase”

p. 3 line 24
replace "ER" by "Endoplasmic Reticulum"

p. 7 line 9
replace "one of them originating from the linker between both domains " by "the latter originating from the linker between the two domains"

p. 9 lines 20-21
"Thr679 of the C-loop is involved in L-Gln hydrolysis (Fig. 1e)". This statement is not illustrated by the Figure.

p. 10 lines 32-33 and p. 11 lines 30-31: I do not think that the hydrogen bond between UDP-GlcNAc and Gln310 is so important because Gln310 is not a conserved residue (Figure S1).

p. 11 line 12
add GDH: "glutamate dehydrogenase GDH"

p. 11 line 14:
the name of GNA-1 should be defined in the main text, not only in the Figure S1 legend.

p. 12 line 17
I suggest to change "significantly reduced" to "reduced" because $k_{cat}(G451E) = 2.2 \text{ s}^{-1}$ instead of $k_{cat}(wt) = 3.6 \text{ s}^{-1}$ is not a big change.

p. 13 line 25
I suggest to change "demonstrating" by "indicating"

p.16 lines 30-32 "reduced reaction rates of mutant G451E in kinetic measurements, compared to wild type GFAT-1. The k_{cat} are rather similar...

Figures

Figure 1e. The H-bond between the α -amino group of Cys2 and Thr94 is not shown.

Figure 2b (monomer B w/o L-Glu and monomer A with L-Glu). The H-bond between Tyr32 and the C-loop is not clear. May be this figure deserves to be shown in stereo.

Figure 5: L-Glu in violet sticks is not shown.

It would be better if the structures were oriented in the same way (eg fig 5a and 5f, fig 4a and 5a). In any case, "synthase" and "glutaminase" should be indicated on the Figures to indicate the positions of the synthase and glutaminase domains for better readability.

Figure S1: The sequence numbering is unreadable because superimposed onto the secondary structure. The 2 phosphorylated serine sites should be highlighted.

To highlight the secondary structure differences between the human and bacterial enzymes (p. 14 lines 23-25: "major differences in extended beta sheets and longer loops connecting alpha helices and beta sheets"), the secondary structure of the E. coli enzyme could be drawn below the sequence alignment.

Legend Figure S3 delete "R-loop" and "C-loop" because only the Q-loop is shown.

Reviewer #2 (Remarks to the Author):

The paper titled "Crystal structure of human GFAT-1 provides a mechanistic basis for the hexosamine pathway activation that modulates protein homeostasis" describes in detail the crystal structure of full length human GFAT-1. The enzyme is a two domain enzyme responsible for the production of UDP-GlcNAc, a key intermediate, used in many regulatory processes. The main emphasis of the authors is on the role in age related diseases, where GlcNAc seems to have proteoprotective functions, as seen in the case of Alzheimers and the Tau-GlcNAcylation. Though isolated structures of domains of the enzyme from bacterial as well as eukaryotic sources are available, this is the first full length structure, which might help to lay the foundations to reveal the suggested interdomain communication thought to be essential for the proper function and regulation of the enzyme.

Overall the manuscript is written well and is a timely and most welcome contribution to our understanding how GlcNAc is interlinked with many different cellular processes. However sometimes the flow of the arguments and the line of thought is not crafted well enough to follow the line of evidence easily.

Anyway, a few minor tweaks and clarifications in some parts of the manuscript and it is more than worthy to be published in Nature Communications

I will outline my comments and suggestions in no particular order below.

Results:

The authors argue about the oligomeric state of the full length protein based on gelfiltration data, crystallographic analysis and DLS. It seems quite convincing to assume a dimeric arrangement based on two of the methods. DLS somewhat seems to point towards a Tetramer, a state observed for the *C. albica* isomerase domain. However, taken the error of a DLS measurement into account, which seems not very convincing. If there is a weakly tetramer, a shoulder or small peak should be visible in the SEC as well. I think SEC-MALLS measurements or SAXS in solution are better suited compared to DLS and should help to clarify this issue.

The authors write alpha and beta as words. I suggest replacing them in the text with the correct symbols as well, for example "three α -helices".

The human isomerase domain consists two.... \diamond The authors use two times "domain". It might be helpful to call them sub-domain or something similar to avoid confusion.

The authors argue about the two states of the respective monomers in regard of the catalytic cycle. It is very hard to draw conclusions, in particular due to the fact that one monomer is locked in its conformation by crystal contacts. That holds true for the conclusion of a sequentially active enzyme. Without the crystal contact both domains might be catalytically independent. In fact there is no evidence that the two monomers depend on each other except sharing an active site residue in the isomerase domain. Nothing seems to be known about further allosteric communication between both monomers.

The authors claim that monomer A are in an intermediate state but with Glu one would rather expect the post catalytic state? Furthermore in both monomers there is no evidence of the ammonia channel, which should form upon binding of both substrates with an ordered C-loop as shown in the *E.coli* structures?

What is with the sugar? Isn't it unusual that the C-loop is completely disordered, taking into account that the C-loop shields the sugar in its binding site? Does it shield the sugar binding site in monomer B?

The binding of UDP-GlcNAc as feedback inhibitor is only characterised by IC50 and the soaking experiment. To get a better picture ITC might be a good technique to compare more quantitatively the two effects of the two mutations on the affinity of GFAT-1 towards UDP-GlcNAc.

Also it would be good to discuss how the binding site compares to known UDP-GlcNAc sites. It seems that the base is not stabilised by aromatic interactions compared to other known binding sites. Could be interesting, if one wants to target this site with small molecules.

The mutation G451E was of particular interest due to its gain of function relevance.

Though there are changes in the k_{cat} for the L-Glu production, comparing the k_{cat}/K_m values for both variants, shows only a change of about two fold which is not very conclusive for a definite judgement of the role of the mutation in the context of the catalytic cycle. The changes in the GlcN6P production are as well compensatory taking k_{cat}/K_m into account. It would be good to add this value to the provided tables in both Figures.

To evaluate the role of the hydrogen bond with Q307 additional mutants were made. Q307A shows a higher sensitivity (lower IC_{50}) towards GlcNAc but not the double mutant (see Supp Fig 5g), at least 10 fold higher IC_{50} compared to wt. The last sentence on page 12 seems to indicate that both mutants behave similar, which doesn't seem right. This needs to be clarified.

In itself the role of Q307 and G451E are not sufficiently clear. It somehow seems that the hydrogen bond has some importance, but Q307A should not have such an effect, especially on FrcgP binding (10 fold higher K_m k_{cat}/K_m change also ten fold) Could there be changes in mobility?

What is meant with asymmetric dimer in the Discussion section on page 14. If that refers to the two conformations observed, I would rather attribute this to the different crystal contacts.

The authors discuss the role of R33 and domain communication upon binding of the substrate. It is not clear from the discussion in its current form if R33 changes may allow domain movement or channel formation? According to the manuscript the sugar binds first followed by the glutamine. So what does the disruption of the salt bridge trigger?

Discussion on page 16 it is discussed that the lack of feedback inhibition is supported by reduced reaction rates. That is true in terms of slower turn over numbers, but the lower K_m compensates for it. The essential question on the role of this modification in the catalytic cycle is still not completely resolved.

The discussion points out that under physiological conditions GFAT-1 might be completely inhibited by cellular HexNAc levels. Is there anything known about the cellular concentration of HexNAc and does UDP-GalNAc show similar feedback inhibition? In previous reports on the bacterial enzyme, changes in the oligomeric state are suggested to be important on activity. Have the authors observed anything in that respect in SEC runs with the different mutants?

Figures:

Figure 2: It might be worth to try cylinders in the overlay for clarity.

Figure 3: In case of subfigure C it might be worth to try if one can remove even more elements to get a clearer figure. The things in the background (sugar) for example can go.

Figure 4: As mentioned above, I would add k_{cat}/K_m for the table.

Figure 6: I recommend to add in the supplementary the full blot to judge the overall quality of the blot.

Response to the Reviewer Comments

Reviewers' comments:

Reviewer #1 (Remarks to the Author):

Ruegenberger et al. report here the first full-length crystal structure at 2.4 Å resolution of human GFAT-11, an important enzyme of the hexosamine pathway. While co-crystallization with the allosteric inhibitor UDP-GlcNAc, which inhibits the glutaminase function, did not lead crystals, UDP-GlcNAc was successfully soaked into the GFAT-1 crystals. However, no structural changes of the glutaminase active site upon binding UDP-GlcNAc were observed, which could have explained feedback inhibition, probably because a larger panel of crystallization conditions needs to be screened to trap a conformation showing how UDP-GlcNAc binding triggers conformational changes in the glutaminase site.

The structures of the G461E and G451E mutants are also reported, together with their kinetic constants and IC50 constants for inhibition by UDP-GlcNAc. Both glycines 461 and 451 are located at the bottom of the UDP-GlcNAc binding pocket. The mutants had kinetic constants similar to those of the wild-type protein and did not respond to UDP-GlcNAc feedback inhibition. While UDP-GlcNAc could not be soaked into the G461E crystals, it could bind to the G451E crystals. The G451E mutation has been shown previously to lead to gain-of-function in *C. elegans* (Denzel et al. *Cell*, 2014) i. e. it leads to elevated hexosamine flux and extended lifespan. Moreover, N2a mouse neuroblastoma cell lines bearing the G451E mutation were shown to be more resistant to tunicamycin and to have an increased level of UDP-N-acetyl-hexosamine compared to cell lines bearing wild-type GFAT-1. As GFAT-1 expression was decreased upon the G451E mutation, the gain-of-function likely results from enhanced GFAT-1 activity.

From their results, the authors conclude that the gain-of-function in the G451E mutant might result from impaired transmission of the UDP-GlcNAc-mediated feedback inhibition from its binding pocket to the site of L-Gln hydrolysis and that the interdomain linker might be involved in signal transmission.

In conclusion, despite this paper reports the first crystal structure of full-length human GFAT-1, which is interesting to a broad scientific audience as activation of GFAT-1 appears to be a promising target to ameliorate proteotoxic diseases, and is technically sound, it does not answer questions about the feedback inhibition mechanism by UDP-GlcNAc, which is the main hallmark of the mammalian enzyme compared to the bacterial enzyme. Although in vivo experiments indicate that point mutations that alter inhibition by the allosteric inhibitor UDP-GlcNAc activate the hexosamine pathway, extending lifespan and limiting ageing-associated pathologies, the structural basis for this activation needs to be consolidated. Therefore I do not recommend publication of this article in the present state by Nature Communications.

We like to thank the reviewer for a differentiated and critical estimation of our work. While the reviewer has pointed to the highlights of our findings, the comments also point to the key question that remained open in the first version of our manuscript in which the UDP-GlcNAc dependent inhibition was not sufficiently explained through

structural changes at the glutaminase active site. With our new experimentation we can now better explain the feedback inhibition.

While one might expect an effect of UDP-GlcNAc on the glutaminase active site conformation, we did not find evidence to support this. However, we carefully analyzed the conservation of GFAT-1 from bacteria to higher organisms. Bacterial GFAT is not inhibited by UDP-GlcNAc while mammalian GFAT-1 is. Thus, regions of the enzyme that differ between prokaryotic and eukaryotic GFAT-1 ought to be informative regarding mechanistic aspects of the feedback inhibition. The interdomain linker showed changes between prokaryotic and eukaryotic GFAT-1. The isomerase and glutaminase active sites of GFAT-1 are coupled and its function depends on the communication and relative orientation of the two domains. Thus, another way to block GFAT's activity might be by interfering with the tight coupling of the active sites by disturbing the orientation of the glutaminase and isomerase domains relative to each other. In this process, the interdomain linker might play a critical role. We show evidence for this hypothesis: First, while the UDP-GlcNAc binding site at the isomerase domain is specific to eukaryotes, the glutaminase active site is fully conserved between bacterial and eukaryotic GFAT (see p. 13 upper part). This makes local changes of the glutaminase site upon UDP-GlcNAc binding unlikely. Second, we demonstrate the importance of the UDP-GlcNAc interaction with the interdomain linker. In support of this, we now included data from an UDP-GalNAc inhibition assay and a structure of UDP-GalNAc-bound GFAT-1 (Fig. 4e,g, Supplementary Fig. 4c-e). UDP-GalNAc interacts with the GFAT isomerase domain but exhibits a weaker inhibitory potential than UDP-GlcNAc due to an altered interaction of the sugar with the interdomain linker. Third, the G451E gain-of-function mutation engages in a new interaction with the interdomain linker, explaining why this mutant, despite binding UDP-GlcNAc, does not show feedback inhibition.

In sum, our data support the idea that UDP-GlcNAc modifies GFAT-1 activity through interactions with the interdomain linker that might affect coupling of GFAT's two enzymatic activities. We re-phrased the discussion and include results from previous studies (Assrir et al. (reference 33), Walter et al. (reference 46)) to support our claim.

Major modifications

A) Oligomerization

p.6 lines 21-28

“and is in accordance with the estimated molecular weight from the gel filtration step”.

The value should be given as well as the theoretical value for the dimer. However, given that GFAT-1 displays an elongated shape, calibration with standards of a SEC column is not enough to determine the molar mass and the oligomerization state of the protein in solution. So I recommend to delete this part of the sentence. The molar mass in solution is given by DLS.

Moreover, I suggest to invert the results in solution and the crystallographic data, as follows:

This dimeric assembly is judged as stable in solution by the PISA server. Yet, the molar mass measured by DLS (323 kDa) indicates a tetrameric structure in solution (monomer molar mass of 78.8 kDa) (Supplementary Fig. 1d). Accordingly, the crystal structure shows that two rather weakly interacting dimers could form a tetramer similar to the one described for the structure of the isolated isomerase domains of the *Candida albicans* ortholog (Supplementary Fig. 1b,c)³⁰. However, this tetramer appears to be less stable than the dimer because the PISA server did not reveal a stable tetramer from crystal packing.

We like to thank the reviewer for this constructive suggestion. As the reviewer, we were concerned about the validity of the analysis of the oligomeric state of GFAT-1, given the contradicting results from SEC and DLS. Since the oligomeric state is not the main focus of this study, we decided to exclude this section from the revised manuscript.

B) Electron density maps

Globally, electron density maps omitting the important regions that are discussed are lacking. Moreover, as part of Nature journals publishing policies for macromolecular crystallographic data, a stereo image of a portion of the electron density map should be provided with the submitted manuscript. None is provided.

We agree that systematically including omit maps strengthens our statements and we have included them where appropriate as discussed point-by-point below. Moreover, we have included Fo-Fc omit maps for the C-loop (Supplementary Fig. 2e), GlcN6P bound to the isomerase site (Supplementary Fig. 2h), UDP-GlcNAc (Supplementary Fig. 4a), and UDP-GalNAc (Supplementary Fig. 4c). In addition, we included a stereo image showing monomer A with bound Glc6P, Glu, and UDP-GlcNAc including the 2Fo-Fc map of all ligands (Fig. 4a). This image helps to get an overview of the localization of the binding sites and their positions on the molecule, which are discussed within the manuscript. In addition, we included stereo images in Supplementary Fig. 1c, 2c, 2d.

p. 7 line 25 “corresponding electron density...” and Figure 1d and 1e

An electron density map omitting Frc6P and Glu contoured at 2.5-3 sigma should be superposed onto the active sites. In particular, it is important to see that the electron density in the isomerase site fits a linear and not a cyclic sugar.

As requested by the reviewer, we generated omit maps for all demanded parts of the structure. The new Supplementary Fig. 1e and h show the omit maps for both active sites. In particular, Supplementary Fig. 1e provides evidence for the presence of a linear sugar in the active site.

p. 7 lines 15-16 and Sup Figure 1F

“S243 was found phosphorylated in both mass spectrometry analysis and the crystal structure (Supplementary Fig. 1a,f).”

An electron density map omitting phosphorylated serine contoured at 2.5-3 sigma should be shown instead of the 2Fo-Fc map ((Supplementary Fig. 1f).

We exchanged the 2Fo-Fc map of phosphorylated Ser243 in supplementary Fig. 1d with a Fo-Fc map omitting the phosphorylated serine (Supplementary Fig. 1d).

p. 8 lines 28-30

“The glutaminase domain of monomer B makes no crystal contacts and is more flexible resulting in partially low electron density. (Supplementary Fig. 2a,b)” No electron density is shown in Sup Figure 2a and 2b.

We show the electron density in Supplementary Fig. 2b and modified this sentence to “the glutaminase domain of monomer B makes no crystal contacts and is more flexible resulting in higher B-factors (Supplementary Fig. 2a) and partially low electron density (Supplementary Fig. 2b).”

p. 11 lines 22-24

“Indeed, UDP-GlcNAc was not bound to the G461E mutant of GFAT-1 after crystal soaking (Fig. 4e)”. A 2Fo-Fc or Fo-Fc map should be shown to illustrate this statement.

In the previous manuscript we included the structure of GFAT-1 G461E with a superposition of UDP-GlcNAc. To address the point of the reviewer, we included the structure of GFAT-1 G461E after an UDP-GlcNAc soaking attempt. Now, we highlight the mutation G461E with a 2Fo-Fc map and demonstrate the absence of UDP-GlcNAc with an Fo-Fc map in Supplementary Fig.4h.

Supp Figure 3.

Moreover, Fo-Fc maps omitting Glu and the Q-loop (and not 2Fo-Fc maps) should be drawn.

Finally, the omit electron density maps should be shown both for the Glu-bound and the Glu-free structures for comparison.

We addressed this comment by including images of Fo-Fc maps omitting Glu (Supplementary Fig. 1h), the Q-loop (Supplementary Fig. 3b), and 6 Å around Glu, both in the Glu-bound and the Glu-free structure (Supplementary Fig. 3a).

Figure 5f. A Fo-Fc omit map should be shown.

The Fo-Fc omit map for UDP-GlcNAc is included in Fig. 5e (old 5f). Moreover, we added a close-up Fo-Fc omit map in Supplementary Fig. 5c.

Concerning the molecule bound at the isomerase site, there is an inconsistency between the text (Glc6P) and the figure legends (Frc6P):

p. 7 lines 30-31 :“we decided to model the product Glc6P in our crystal structures”
figure legends (eg p. 31 line 29: “Frc6P (yellow sticks)”

Thanks for pointing out this mistake, it is corrected in the figure legends.

C) Catalytic states of the structures

Discussion p. 14 lines 4-5

“In contrast to bacterial GFAT, human GFAT-1 forms an asymmetric dimer.” This sentence should be omitted. Indeed, it is discussed below (lines 29 and following) and I don't think that it is a peculiarity of the human enzyme compared to the bacterial enzyme. I think that it is solely a different crystal packing that has frozen two conformational states for the monomer in the human enzyme, and not in the bacterial enzyme.

We thank the reviewer for pointing this out, the discussion is modified accordingly.

“For the first time, we observed a specific interaction between the two GFAT-1 domains relevant for the orchestrated coupling of the catalytic cycle.”

It is not clear to me what is this specific interaction... Explain or delete

We deleted this statement.

1) p. 9 lines 23-29

“Although both active sites are occupied by ligands in monomer A, no ammonia channel connecting the glutaminase and the isomerase site is formed.”

“Thus, the conformation of monomer A might be a snapshot of the intermediate state after binding of both substrates and before ammonia channel formation.”

2) p. 14 lines 31-34

“In monomer A, L-Glu was bound to GFAT-1, which was associated with conformational changes that would not allow GlcN6P synthesis.”

3) and p. 15 lines 2-4:

“This is in contrast to reports from the *E. coli* enzyme, where a rotation of Trp96 equivalent Trp74 opens the ammonia channel that is mainly formed by the C-loop17. These data suggest that L-Glu binding induces changes similar to L-Gln binding, but inactivates Cys2 and does not provoke the formation of the ammonia channel. Hence, the catalytic cycle cannot be completed and GFAT-1 is fixed in this state”.

These statements are wrong. There is a confusion between Glutamine (the substrate) and Glutamate (the product). In the *E. coli* structure (reference 17), the competitive inhibitor DON, an analog of the glutamine substrate (and not the Glu product) occupies the glutaminase active site and the Frc6P substrate occupies the isomerase site. Thus, the DON/Frc6P *E. coli* structure mimics a catalytic intermediate state of the enzyme, in which the ammonia channel is open. In contrast, in the human structure, L-Glu is the product and does NOT induce changes similar to substrate (L-Gln) binding. According to me, GFAT1 with Frc6P or Glc6P bound at the isomerase site and the L-Glu product bound at the glutaminase site does not correspond to a catalytic state.

After careful reevaluation of our data, we agree with the reviewer and are grateful for pointing out this weakness in our interpretation. We have now removed the passages referring to the catalytic cycle.

“This indicates that the two monomers might be sequentially active”, NO
“meaning that the second monomer can bind to L-Gln only when the other monomer has completed its catalytic cycle.” NO: “meaning that in the second monomer the L-Glu product has already left the active site.”

The reviewer is correct, the asymmetric dimer in our structure does not provide evidence for the conclusions. We have re-phrased the results and discussion section accordingly (see p. 8 (middle), p. 15f (bottom)). In the new version of our manuscript, we describe the two monomers A and B and compare them to known structures of *E. coli*.

p. 10 lines 18-19

“Together, the glutamate-free structure of monomer A suggests a communication of the glutaminase site through Arg33 with the isomerase domain, observed for the first time.”

“observed for the first time” is wrong. It should be replaced by “as observed for *E. coli* GImS” Indeed, in Reference 17, it was already shown that in *E. coli* GImS, Arg26 (equivalent to Arg33 in GFAT-1) together with the C-terminal residues of the isomerase domain are major actors of the communication between the two active sites.

“ The communication between the two active sites is mediated by residues forming part of the channel, Arg-26 and the C-terminal residues including Thr-606” (last lines of the article).

We agree that our statement was misleading and thank the reviewer for pointing this out. We are aware that Arg33 (in *E. coli* Arg26) was previously suggested to be important in interdomain communication. In their study, Mouilleron et al. (reference 17) compared the structure of *E. coli* GImS with Frc6P with the structure where both active sites are occupied by DON and Frc6P (see above). They nicely describe the closure of the Q-loop, as well as the formation and opening of the ammonia channel upon DON binding. However, in their study, *E. coli* Arg26 shows in both the Frc6P-bound and the Frc6P+DON-bound structures interactions within the glutaminase domain and the flexible C-loop (see images “orientation of Arg33”). In our study, we show for the first

time an interaction of human Arg33 with Asp667 of the isomerase domain (see images “orientation of Arg33”). Prompted by the reviewer’s comment we deleted the statement “observed for the first time“, but decided to not add “as observed for *E. coli* GlmS”, because this would be misleading. In fact, we did not observe the same Arg33 orientation as it is reported for the *E. coli* homolog. To further address the reviewer’s comment, we have re-phrased the discussion (see p. 16 (bottom)).

Orientation of Arg33 (*E. coli* Arg26) in the presence or absence of DON or L-Glu. Numbering according the human sequence.

p. 10 lines 24-25

“Co-crystallization of GFAT-1 with UDP-GlcNAc did not yield any crystals.”

It should be specified if only crystallization has been tried around the standard condition (0.1 M Bis tris propane pH 8.5, 0.2 M Potassium sodium tartrate and 20 % (w/v) PEG3350) or if a whole range of different conditions using various crystallization kits have been spanned.

Indeed, as part of the revision process, we tested a whole range of different crystallization conditions screening for GFAT-1 + UDP-GlcNAc co-crystallization conditions and obtained poor-diffracting crystals. Any further optimization and post-crystallization treatments (annealing and dehydration) did not help to improve the resolution beyond 4.5-5.0 Å. In these datasets, we found a very large unit cell with an estimated 12 GFAT molecules, according to Matthews coefficient. However, we could only place five GFAT-1 molecules into the density using monomer B as search model. The isomerase domains were a good fit in the density and we even could see positive difference density at the UDP-GlcNAc binding sites indicating the presence of UDP-GlcNAc. For the glutaminase domain, we found at least three different orientations relative to their respective isomerase domain (two glutaminase domains matched the

orientation of both domains that we saw in monomer B). However, we were not able to place additional GFAT-1 molecules in the density and could not refine the structure further. We are happy to share these data upon request.

We changed the statement to: “Co-crystallization of GFAT-1 with UDP-GlcNAc did not yield well-diffracting crystals.”

p. 10 lines 26-28 and p. 32 line 23 Legend Figure 4.

An overview figure showing the respective location of Glu6P and UDP-NAc-GlcN6P in the isomerase domain is lacking. Also, it is not clear if Glc6P or Frc6P is present in UDP-NAc-GlcN6P-bound GFAT-1: Frc6P in yellow sticks is not seen in Figure 4. If present, change the orientation or show a stereo figure!

To address the reviewer’s comment, we added another structure in our manuscript (GFAT-1 WT +Glu +Glc6P +UDPGlcNAc) and show a stereo image of GFAT-1 with all ligands bound in Figure 4a. Unfortunately, at the given resolution we cannot distinguish between Glc6P or Frc6P in our structures. On the middle of page 7 we explain why we decided to build Glc6P into the structures.

p. 13 line 5-6

“Furthermore, the interdomain linker might be involved in signal transmission.“ I think this statement is an over-interpretation of the crystallographic data.

We agree that the interdomain linker does not signal to the glutaminase active site (see above). We have re-phrased the discussion accordingly (see p. 17f).

Minor modifications

Abstract line 1 and p. 4 line 2

“amidotransferase” and not “aminotransferase”

We implemented this suggestion in our manuscript.

p. 3 line 24

replace “ER” by “Endoplasmic Reticulum”

We implemented this suggestion in our manuscript.

p. 7 line 9

replace “one of them originating from the linker between both domains ” by “the latter originating from the linker between the two domains”

We implemented this suggestion in our manuscript.

p. 9 lines 20-21

“Thr679 of the C-loop is involved in L-Gln hydrolysis (Fig. 1e)”. This statement is not illustrated by the Figure.

The reviewer is correct, Figure 1e does not support this statement and we deleted the reference to Figure 1e. However, in the section “GFAT-1 active sites are conserved from bacteria to humans” we discuss the high conservation of all catalytically relevant residues including Thr679. Since the C-loop is a very flexible part of the structure, Thr679 can take part in catalysis although this is not supported by Figure 1e.

p. 10 lines 32-33 and p. 11 lines 30-31: I do not think that the hydrogen bond between UDP-GlcNAc and Gln310 is so important because Gln310 is not a conserved residue (Figure S1).

We agree with the reviewer that these data do not add substantially to our conclusions and have therefore removed them from the manuscript.

p. 11 line 12

add GDH: “glutamate dehydrogenase GDH”

We implemented this suggestion in our manuscript.

p. 11 line 14:

the name of GNA-1 should be defined in the main text, not only in the Figure S1 legend.

We implemented this suggestion in our manuscript.

p. 12 line 17

I suggest to change “significantly reduced” to “reduced” because k_{cat} (G451E)=2.2 s⁻¹ instead of $k_{cat}(wt) = 3.6$ s⁻¹ is not a big change.

We deleted this statement and included k_{cat}/K_m values in our analyses for a comparison of wild type and G451E GFAT-1 (see p. 12 (middle)).

p. 13 line 25

I suggest to change “demonstrating” by “indicating”

We implemented this suggestion in our manuscript.

p.16 lines 30-32 “reduced reaction rates of mutant G451E in kinetic measurements, compared to wild type GFAT-1. The k_{cat} are rather similar...”

The reviewer is correct; therefore, we deleted this statement.

Figures

Figure 1e. The H-bond between the γ -amino group of Cys2 and Thr94 is not shown.

We added this H-bond in our figure. However, we have to admit that it is hardly visible. Since this interaction is not of great importance and a change of the orientation would affect the visibility of the other interactions, we decided to stay with the current view.

Figure 2b (monomer B w/o L-Glu and monomer A with L-Glu). The H-bond between Tyr32 and the C-loop is not clear. Maybe this figure deserves to be shown in stereo.

Prompted by the reviewer’s comment we generated stereo images for monomer B w/o L-Glu and monomer A with L-Glu. However, in the stereo images we lost the direct comparison of the Glu-free and Glu-bound states side-by-side. Therefore, we included an enlarged version of Fig. 2b in the main figures and added the stereo images of monomer B w/o L-Glu and monomer A with L-Glu in the Supplements (Supplementary Fig. 2c,d).

Figure 5: L-Glu in violet sticks is not shown.

It would be better if the structures were oriented in the same way (eg fig 5a and 5f, fig 4a and 5a). In any case, “synthase” and “glutaminase” should be indicated on the Figures to indicate the positions of the synthase and glutaminase domains for better readability.

Indeed, L-Glu was not visible at the given orientation. Now, we addressed the reviewers’ comment and orientated most overview structures (Fig 4b, 5a, 5e, and 6a) in the same way and added the labels “isomerase” and “glutaminase”.

Figure S1: The sequence numbering is unreadable because superimposed onto the secondary structure. The 2 phosphorylated serine sites should be highlighted. To highlight the secondary structure differences between the human and bacterial enzymes (p. 14 lines 23-25: “major differences in extended beta sheets and longer loops connecting alpha helices and beta sheets”), the secondary structure of the *E. coli* enzyme could be drawn below the sequence alignment.

We changed the position of the secondary structure elements to ensure a good readability of the sequence numbering and added the known phosphorylation sites. Since the human and bacterial enzymes differ only in extended beta sheets and loops, we highlighted these regions with yellow bars. Moreover, we added a stereo image of the superposition of the human and bacterial glutaminase domain in Supplementary Figure 1c.

Legend Figure S3 delete “R-loop” and “C-loop” because only the Q-loop is shown.

Supplemental Figure 3 supports conclusions that were drawn for Figure 3 (“Glutamate determines conformation of Q-loop, R-loop, and C-loop”). Since both figures are related, we decided to stay with the same title for main and supplemental figures, even though the R- and C-loop do not occur in the supplemental figure. We will confer with the editor to make sure the figures are labelled properly.

Reviewer #2 (Remarks to the Author):

The paper titled “Crystal structure of human GFAT-1 provides a mechanistic basis for the hexosamine pathway activation that modulates protein homeostasis” describes in detail the crystal structure of full length human GFAT-1. The enzyme is a two domain enzyme responsible for the production of UDP-GlcNAc, a key intermediate, used in many regulatory processes. The main emphasis of the authors is on the role in age related diseases, where GlcNAc seems to have proteoprotective functions, as seen in the case of Alzheimers and the Tau-GlcNAcylation. Though isolated structures of domains of the enzyme from bacterial as well as eukaryotic sources are available, this is the first full length structure, which might help to lay the foundations to reveal the suggested interdomain communication thought to be essential for the proper function and regulation of the enzyme.

Overall the manuscript is written well and is a timely and most welcome contribution to our understanding how GlcNAc is interlinked with many different cellular processes. However sometimes the flow of the arguments and the line of thought is not crafted well enough to follow the line of evidence easily.

Anyway, a few minor tweaks and clarifications in some parts of the manuscript and it is more than worthy to be published in Nature Communications

We thank the reviewer for the enthusiasm for our manuscript and appreciate the comments aimed to improve the paper. We will address them point-by-point below.

I will outline my comments and suggestions in no particular order below.

Results:

The authors argue about the oligomeric state of the full length protein based on gelfiltration data, crystallographic analysis and DLS. It seems quite convincing to assume a dimeric arrangement based on two of the methods. DLS somewhat seems to point towards a Tetramer, a state observed for the *C. albica* isomerase domain. However, taken the error of a DLS measurement into account, which seems not very convincing. If there is a weakly tetramer, a shoulder or small peak should be visible in the SEC as well. I think SEC-MALLS measurements or SAXS in solution are better suited compared to DLS and should help to clarify this issue.

We agree with the reviewer that SEC-MALLS or SAXS would have been perfect methods to support our data on the oligomeric state of GFAT-1. However, we decided to exclude the contradicting results regarding the oligomeric state, because the oligomeric state of GFAT-1 was not main focus of this study.

The authors write alpha and beta as words. I suggest replacing them in the text with the correct symbol as well, for example “three α -helices”.

The human isomerase domain consists two..... \diamond The authors use two times “domain”. It might be helpful to call them sub-domain or something similar to avoid confusion.

We implemented these changes and replaced the words “alpha” and “beta” by their symbols and renamed the second “domain” from the SIS domain “sub-domain”.

The authors argue about the two states of the respective monomers in regard of the catalytic cycle. It is very hard to draw conclusions, in particular due to the fact that one monomer is locked in its conformation by crystal contacts. That holds true for the conclusion of a sequentially active enzyme. Without the crystal contact both domains might be catalytically independent. In fact there is no evidence that the two monomers depend on each other except sharing an active site residue in the isomerase domain. Nothing seems to be known about further allosteric

communication between both monomers.

The authors claim that monomer A are in an intermediate state but with Glu one would rather expect the post catalytic state? Furthermore in both monomers there is no evidence of the ammonia channel, which should form upon binding of both substrates with an ordered C-loop as shown in the *E. coli* structures?

We concur with the reviewer that we leaned too heavily on the asymmetry of our GFAT-1 dimer in our first version of the manuscript. Therefore, we re-phrased the results and discussion describing the two monomers A and B with their ligands and compare them with the known *E. coli* structures. Motivated by the reviewer's comment, we co-crystallized GFAT-1 with its two products L-Glu and D-GlcN6P to further analyze the post catalytic state. As shown in the new Supplemental Figures 2f-h, we found a linear sugar in the active site and see the same asymmetric dimer in our structure as for the Glc6P-bound structure. The description of these results has been added on page 9 (middle). We did not observe the formation of the ammonia channel in either monomer because in monomer B the glutaminase site was not occupied by any ligand and in monomer A the product L-Glu was bound to the glutaminase site. Obviously, Glu-binding was insufficient to induce the ammonia channel formation.

What is with the sugar? Isn't it unusual that the C-loop is completely disordered, taking into account that the C-loop shields the sugar in its binding site? Does it shield the sugar binding site in monomer B?

In our structures, we found the C-loop shielding the sugar binding site in monomer B. There, its position is fixed by interactions with the R-loop and Q-loop from the glutaminase domain (Fig. 2b, see p. 8 (middle)). In contrast, in monomer A the C-loop adopts a completely different conformation. Due to crystal contacts, the glutaminase and isomerase domains of monomer A are in a more open conformation. Thereby, the C-loop loses the stabilizing interactions with the R- and Q-loops (Fig. 2b, see p. 8 (bottom)). Thus, the C-loop seems to need a stabilization by interactions with the glutaminase domain in order to ensure proper covering of the sugar binding site.

The binding of UDP-GlcNAc as feedback inhibitor is only characterised by IC50 and the soaking experiment. To get a better picture ITC might be a good technique to compare more quantitatively the two effects of the two mutations on the affinity of GFAT-1 towards UDP-GlcNAc.

We fully agree that a detailed characterization of binding affinities would have further strengthened our manuscript. However, any detailed study using ITC would cause a substantial delay.

Also it would be good to discuss how the binding site compares to known UDP-GlcNAc sites. It seems that the base is not stabilised by aromatic interactions compared to other known binding sites. Could be interesting, if one wants to target this site with small molecules.

Assrir et al. (reference 33) reported that UDP-GlcNAc, UDP-Glc, and UDP are able to bind with similar K_D values to GFAT-1. Therefore, the UDP-moiety seems to be sufficient to mediate binding. This is in line with our results, that despite the hydrogen bonds from the sugar to Gln310, all other side chains interact with the UDP-moiety. Other human UDP-GlcNAc binding sites, which are described for example for O-GlcNAc Transferase (PDB 4GZ5), UDP-GlcNAc pyrophosphorylase (PDB 1JV1), or UDP-galactose 4-epimerase (PDB 1HZJ) show more interactions to the GlcNAc and seem to be more specific.

The mutation G451E was of particular interest due to its gain of function relevance. Though there are changes in the k_{cat} for the L-Glu production, comparing the k_{cat}/K_m values for both variants, shows only a change of about two fold which is not very conclusive for a definite judgement of the role of the mutation in the context of the catalytic cycle. The changes in the GlcN6P production are as well compensatory taking k_{cat}/K_m into account. It would be good to add this value to the provided tables in both Figures.

Prompted by the reviewer's comment we included k_{cat}/K_m values in our tables and included them in the judgement of the catalytic efficiency of our mutant G451E (see p. 12 (middle)). The reviewer is correct, with a k_{cat}/K_m of 21.3 for the wild type and a k_{cat}/K_m of 20.0 for mutant G451E, there is no effect of the mutation on the synthesis of the product D-GlcN6P.

To evaluate the role of the hydrogen bond with Q307 additional mutants were made. Q307A shows a higher sensitivity (lower IC₅₀) towards GlcNAc but not the double mutant (see Supp Fig 5g), at least 10 fold higher IC₅₀ compared to wt. The last sentence on page 12 seems to indicate that both mutants behave similar, which doesn't seem right. This needs to be clarified.

In itself the role of Q307 and G451E are not sufficiently clear. It somehow seems that the hydrogen bond has some importance, but Q307A should not have such an effect, especially on FrcgP binding (10 fold higher K_m k_{cat}/K_m change also ten fold) Could there be changes in mobility?

We re-phrased our results in order to emphasize how each mutant reacted to UDP-GlcNAc inhibition (see p. 12 (bottom)). Furthermore, we included potential explanations for the different sensitivities in the discussion (see p. 17 (bottom)). In summary, we think that the mutations interfere with the flexibility of the interdomain linker and the positioning of the interdomain linker relative to the isomerase domain/UDP-GlcNAc binding site. In G451E, the additional salt bridge stabilizes the linker and the two side chains (Glu451, Gln307) prevent a close interaction. The mutation Q307A might increase the flexibility of the interdomain linker, and the smaller Ala side chain allows a close interaction of interdomain linker and UDP-GlcNAc-bound isomerase domain. In Q307A/G451E the interdomain linker is still flexible, but the Glu451 side chain might prevent the correct interactions of the interdomain linker with the UDP-GlcNAc-bound isomerase domain.

What is meant with asymmetric dimer in the Discussion section on page 14. If that refers to the two conformations observed, I would rather attribute this to the different crystal contacts.

We thank the reviewer for pointing this out; the asymmetric dimer seems to be a crystallographic artefact. Therefore, we re-phrased the discussion omitting the previous interpretation of the two conformational states found within the asymmetric dimer.

The authors discuss the role of R33 and domain communication upon binding of the substrate. It is not clear from the discussion in its current form if R33 changes may allow domain movement or channel formation? According to the manuscript the sugar binds first followed by the glutamine. So what does the disruption of the salt bridge trigger?

For the *E. coli* enzyme, it is reported that Frc6P-binding triggers a hinge movement of the C-loop and DON-binding induces a rotation of the glutaminase domain (reference 17). However, in our crystal system such a re-orientation of the glutaminase domain is

restricted by crystal contacts. At this point, we have to speculate as to the consequences of the disruption of the salt bridge. The Arg33-Asp667 salt bridge might keep the cleft between isomerase and glutaminase domain open. L-Glu binding induces the Q-loop closure and subsequently the R-loop flip, disrupting the salt bridge. Thereby, both domains could get in close contact and the C-loop can interact with both active sites.

Discussion on page 16 it is discussed that the lack of feedback inhibition is supported by reduced reaction rates. That is true in terms of slower turn over numbers, but the lower K_m compensates for it. The essential question on the role of this modification in the catalytic cycle is still not completely resolved.

We agree with the reviewer that the lower K_m compensates for the lower k_{cat} . The addition of k_{cat}/K_m values in our results make this clearer (see above). Therefore, we deleted the statement that we see reduced reaction rates. Interestingly, the k_{cat}/K_m values for Glu-production correlate with the sensitivity to UDP-GlcNAc (higher k_{cat}/K_m = higher sensitivity). We speculate that UDP-GlcNAc might modulate the relative positioning of the two domains and the interdomain linker might be involved in this process. Thus, mutations at the linker region might also interfere with catalysis. However, overall we detect no more than two-fold differences in k_{cat}/K_m and these small changes remain inconclusive.

The discussion points out that under physiological conditions GFAT-1 might be completely inhibited by cellular HexNAc levels. Is there anything known about the cellular concentration of HexNAc and does UDP-GalNAc show similar feedback inhibition?

This is an excellent comment that we should have thought of earlier. We now tested the effect of UDP-GalNAc on the activity of GFAT-1 and found an inhibition of GFAT-1 at only very high doses (new Fig. 4g). We also determined the crystal structure of GFAT-1 with UDP-GalNAc, which gave interesting insight into a potential role of the interdomain linker in feedback inhibition (new Fig. 4e). We have included these findings in the results section (structure: see p. 10 (bottom), inhibition: see p. 11 (bottom)). With regard to the physiological concentrations of UDP-GlcNAc, Park et al. (Nat Chem Biol. 2016 Jul;12(7):482-9) found a concentration of approximately 9 mM in mammalian iBMK cells. This would suggest that GFAT-1 would be fully inhibited under physiological conditions.

In previous reports on the bacterial enzyme, changes in the oligomeric state are suggested to be important on activity. Have the authors observed anything in that respect in SEC runs with the different mutants?

We found all GFAT-1 variant as dimers in our SEC runs. Interestingly, for the wild type enzyme we observed a concentration-dependent shift from dimer to tetramer in our first DLS measurement. This concentration dependent oligomerization was reported for the *E. coli* enzyme. However, despite this first measurement we could never again find a dimer in our DLS measurement even at low concentrations. So, there might be a similar effect, which might be interesting to study in future.

Figures:

Figure 2: It might be worth to try cylinders in the overlay for clarity.

Figure 3: In case of subfigure C it might be worth to try if one can remove even more elements to get a clearer figure. The things in the background (sugar) for example can go.

Figure 4: As mentioned above, I would add k_{cat}/K_m for the table.

Figure 6: I recommend to add in the supplementary the full blot to judge the overall quality of the blot.

We implemented all these nice suggestions in our figures.

Reviewers' comments:

Reviewer #1 (Remarks to the Author):

Several criticisms have been addressed in the revised version of the manuscript.

Moreover, two new structures of GFAT-1 in complex with Glu and GlcN6P, and in complex with UDP-GalNac have been added.

However, the main goal of the manuscript, understand hexosamine pathway regulation at the molecular level (1st line p.6) and, in particular, explain how the G451E gain-of-function mutant (which results in increased cellular UDP-GlcNac levels) acts at the molecular level is not reached. So the title "Crystal structure of human GFAT-1 PROVIDES A MECHANISTIC BASIS for hexosamine pathway activation that modulates protein homeostasis" does NOT reflect the content of the manuscript.

Indeed, the lack of feedback inhibition of UDP-GlcNac (values in figure 5e) of the mutant is not explained by the crystal structure because soaking experiments show that the molecule can still bind to the mutated protein. Moreover the catalytic constants of the G451E mutant ($k_{cat}/K_m = 1.7 \text{ s}^{-1} \text{ mM}^{-1}$ for L-Glu production and $20 \text{ s}^{-1} \text{ mM}^{-1}$ for D-GlcN6P production) are similar to those of the wild-type enzyme (resp 3.3 and $21.25 \text{ s}^{-1} \text{ mM}^{-1}$). Thus the catalytic constants of the mutant do not explain why the gain-of-function could be linked to a higher synthesis of GlcN6P and UDP-GlcNac. The authors agree that these small changes remain inconclusive (response to Referee 2, Discussion on page 16). As a consequence, the discussion deals with speculations about the mechanism of the G451E mutant and the role of the linker region, which are not comforted by experimental results.

More experiments are needed to understand HP regulation at the molecular level, such as determination of the dissociation constants of UDP-GlcNac with wild type and the G451E mutant as well as getting a crystal structure of the G451E mutant co-crystallized with UDPGlcNac, in which the glutaminase domains are not involved in crystal contacts.

Moreover, the structure in complex with Glu has not been correctly revisited with the view that Glu is the product and not the substrate.

I again do not support the publication of this article in Nature Communications because I think that, although the GFAT-1 structure remains interesting as the first structure of the human enzyme, this paper is not a story about elucidating the interdomain communication thought to be essential for the regulation of the enzyme and even less about understanding the mechanism of proteoprotective function of gain-of-function mutants.

According to me, this paper tells a story about the conformational changes occurring during catalysis upon departure of the glutamate product (comparison of the structures of GFAT-1 in complex with Glu and Glc6P/GlcN6P or in complex with Glc6P/GlcN6P alone). Another important thing that has been shown is that the interdomain linker has a role in UDP-GlcNac inhibition. However, UDP-GlcNac binding did not lead to structural changes at the glutaminase site, thus giving no explanation about the transmission of the inhibitory signal to the glutaminase active site.

Major points

Abstract

delete "owing to different interdomain linker contacts". This does not explain why there is no inhibition of the catalytic activity. As stated above, nothing in the crystallographic results explains why UDP-GlcNac binds to GFAT-1 whereas there is a loss of UDP-GlcNac feedback inhibition.

p. 4 line 24

after "Most insights into GFAT structure come from its bacterial homolog, glucosamine-6-phosphate synthase (GlmS)" add "whose full-length structure has been determined in complex with substrates and product". Omitting this statement can lead to misleading understanding that only structures of the isolated domains of the bacterial enzyme have

been solved (see Reviewer 2, 1st comment: Though isolated structures of domains of the enzyme from bacterial as well as eukaryotic sources are available, this is the first full length structure...)

P. 5 lines 4-6. "Structural and functional analyses of point mutants show that their gain-of function results from loss of UDP-GlcNAc inhibition." This statement is wrong because the structural analysis of point mutant does not explain why the gain-of-function comes from loss of UDP-GlcNAc inhibition.

p. 8 lines 16-18: "This open conformation in monomer A allows conformational changes at the active sites with loop movements, which are restricted in the closed conformation in monomer B." This is wrong to say that the open state of monomer A allows for conformational changes that cannot occur in monomer B because the conformation of monomer A (not B) is restricted by crystallographic contacts. The conformation of monomer A thus may represent a non-catalytically relevant conformational state.

p. 8 lines 25-28: "Compared to the E. coli structures, the conformation of monomer B is consistent with the first steps of the catalytic cycle, comprising initial sugar binding and structural ordering of Cloop and glutaminase domain(40)". Or rather the B monomer, in complex with Glc6P, corresponds the last step of catalysis, just before departure of the Glc6P product, and is more like the E. coli GlmS/GlcN6P structure (reference 45)? And the A monomer with Glu and Glc6P bound corresponds to the preceding step, just before the Glu product has left? In that case, it is Glu departure (and not "L-Glu-induced shifts" p. 8 before last line) that triggers conformational changes of the Q and R loops. My proposal is supported by the new structure in complex with Glu and GlcN6P which is very similar to the structure in complex with Glu and Glc6P.

p.9 lines 9-16 "These data suggest that the Q-loop covers the glutaminase active site upon LGlu binding, but as it is the product, it cannot induce the coupling of both active sites by ammonia channel formation." No. It means that after reaction has occurred (Glc6P/GlcN6P formation), the channel gets closed.

Replace "We wondered how the presence of L-Glu and GlcN6P would affect the structure and co-crystallized GFAT-1 in the presence of both product molecules." by "The proposal that monomer A of the GFAT-1 structure in complex with Glu and Glc6P mimics GFAT-1 in complex with both products is in agreement with the structure of GFAT-1 in complex with Glu and GlcN6P..."

p. 9 line 18: change "Given the apparent role of L-Glu in the conformation of the Q- and Rloops, we wondered how its absence would affect the structure of GFAT-1 monomer A

" by "Given that the occupancy of the glutaminase site by Glu is linked to the conformation of the Q- and R-loops, we wanted to know the conformation of these loops in the absence of Glu"

p. 10 lines 4-7 I suggest to change "Together, the glutamate-free structure of monomer A suggests a communication of the glutaminase site through Arg33 with the isomerase domain." by something like "the comparison of the structures of GFAT-1 in complex with Glu and Glc6P/GlcN6P or in complex with Glc6P/GlcN6P alone shows that the signal of glutamate departure is communicated to the isomerase domain through Arg33".

p. 13 line 15 delete "Understanding GFAT-1 gain-of-function by a specific point mutation"

p. 15 Lines 3-4 "present new mechanistic insights into its regulation that affects protein homeostasis though the HP." I disagree.

Legend Figure S3. Change "Glutamate determines conformation of Q-loop, R-loop, and C-loop" by "The conformations of Q-loop, R-loop, and C-loop are linked to the occupancy of the glutaminase site by Glutamate"

p. 16 lines 4-8 "In contrast, no ammonia channel was formed after L-Glu binding in our structure. These data suggest that L-Glu binding induces some changes similar to L-Gln binding, but as it is the product and not substrate of the reaction, it switches Cys2 to a catalytic incompetent rotamer and does not provoke the formation of the ammonia channel." My interpretation is that the structure with the Glu product cannot be compared with the structure with the Gln substrate. Same comment as p. 8 lines 25-28.

p. 16 lines 26-30 "After disruption of the salt bridge by reorientation of the R-loop upon glutamate binding, the cleft could close. This allows the C-loop to interact with both active sites,

which is necessary for catalysis. Thereby, the R-loop might signal the presence of substrate in the glutaminase active site to the isomerase domain." I would tell the story in the other way, following the catalytic cycle: which conformational changes occur upon glutamate release. I do not believe that the structure with glutamate mimics the structure with glutamine. But what seems interesting is that the authors have shown for the first time that the salt bridge between Arg33 and Asp262 breaks when glutamate leaves and a new salt bridge occurs between Arg33 and Asp667 (from the isomerase domain), perhaps preparing the release of the Glc6P/GlcN6P product?

Discussion p. 17 and 18 "There are several lines of evidence that support a key role of the interdomain linker in this process". The discussion is very confused because it mixes arguments that indicate that UDP-GlcNAc could function as a feedback inhibitor by altering the conformation of the linker region and thus the communication between the two active sites (which is probable) and arguments to try to explain why the feedback inhibition would be released in the G451E mutant (for which there is no mechanism that can be drawn from the present study). Also, the increased sensitivity of the Q307A mutant to UDP-GlcNAc compared to the wild-type protein is very difficult to understand at present. An important role of Q307 seems unlikely given it is not conserved (Lys in *C. elegans*, Thr in *D. melanogaster*, Ser in *A. thaliana* and *V. radiata* Figure 6B). p. 17 Lines 30-32 "Furthermore, in the double mutant, Glu451 might prevent the correct positioning of interdomain linker and UDP-GlcNAc-bound isomerase domain". Not understood

Minor points:

p. 6 line 17: delete "as" in "form as an asymmetric dimer"

p. 7 line 11: replace "which was formed " by "which is formed"

p. 8 lines 7-10: the sentence "The two crystallographically independent monomers form an asymmetric dimer due to the shift of the glutaminase domains relative to their respective isomerase moieties, which themselves form a symmetric assembly" is not clear. Replace by "The two crystallographically independent monomers form a dimer. The dimer is asymmetric because the two glutaminase domains are oriented differently relative to their respective isomerase moieties, which form a symmetric assembly".

p.9 before last line replace "rotates" by "is rotated"

p. 10 give the name of GalNAc

p. 11 5th line replace "established" by "used" because these methods have already been published.

It would be clearer if all the catalytic constants, including k_{cat}/K_m are given in a main Table (rather than in Fig and Supp Figures). Then, the numbers need not to be given in the main text.

p. 11 Supplementary Figure 4h does not exist.

p. 11 last lines. Replace "our data support a critical functional role of the sugar's 4-OH interaction with the interdomain linker in the feedback inhibition." by "feedback inhibition of UDP-GlcNAc involves a critical interaction between the sugar 3OH group and the interdomain linker."

Reviewer #2 (Remarks to the Author):

The manuscript entitled „Crystal structure of human GFAT-1 provides mechanistic basis for hexosamine pathways activation that modulates proteostasis" is a resubmission after revision. The authors did a tremendous amount of work to address the comments of the reviewers and improved the manuscript including additional experiments to support their claim. With a few minor tweaks, the manuscript will be more than ready for publication

Title:

I feel, there should be an "a" before mechanistic.

Introduction

Page 4:

Most insights into the GFAT structure and function.....

Page 5:

Interfering with GFAT-1..... The authors speak about GFAT in general and specify here to GFAT-1. Why?

Page 10

There are only two interactions of the glucose ... \diamond N-Acetylglucosamine

Page 12

The authors say that in both, wt and mutant G451E UDP-GlcNAc (5 mM) could be found into the active site, though G451E is not inhibited by UDP-GlcNAc. Is that true for 5 mM in solution?

Page 14

Is it really enhance activity of GFAT-1 or rather constant activity due to the lack of feedback inhibition? The kinetic data suggest the latter.

Page 16

Reorientation of the R-loop upon glutamine

Page 17 and 18

The authors cite Assrir et al. that UDP or UDP-Glc does not inhibit GFAT-1. Taken into account that the changes upon binding or not near the sugar, it makes sense to observe the changes for GlcNAc and GalNAc, emphasizes that the UDP moiety is only for binding and maybe orientation of the HexNAc unit.

Is there anything visible from the structures (changes in the loops), interactions etc., which could hint to a possible cause of the interactions with the sugar part? Q307 was suggested to interact with the HexNAc unit, as far as I understood.

From previous pages I got the impression that the correct orientation and tight coupling is necessary for efficient catalysis for example the formation of the ammonia channel. But on page 17 the authors say that Q307 and E451 prevent close interactions. That sounds contradictory. I would expect that both residues stabilize a closer conformation and Q307A might even allow for a closer and tighter conformation?

Figures.

Fig2 a-e) (according to legend) shows only panels a-d

Fig. 4 What is the role of the shown Magnesium? It seems too far away to be important for the interaction with the nucleotide?

Fig 4c) This is not essential, but to engage in two H bonds Q310 should have a switched side chain. Might be supported by B-factors?

Supplementary Material:

Supp Fig1 exists twice (numbering issue)

Reviewers' comments:

Reviewer #1 (Remarks to the Author):

Several criticisms have been addressed in the revised version of the manuscript. Moreover, two new structures of GFAT-1 in complex with Glu and GlcN6P, and in complex with UDP-GalNAc have been added. However, the main goal of the manuscript, understand hexosamine pathway regulation at the molecular level (1st line p.6) and, in particular, explain how the G451E gain-of-function mutant (which results in increased cellular UDP-GlcNAc levels) acts at the molecular level is not reached. So the title “Crystal structure of human GFAT-1 PROVIDES A MECHANISTIC BASIS for hexosamine pathway activation that modulates protein homeostasis” does NOT reflect the content of the manuscript. Indeed, the lack of feedback inhibition of UDP-GlcNAc (values in figure 5e) of the mutant is not explained by the crystal structure because soaking experiments show that the molecule can still bind to the mutated protein. Moreover the catalytic constants of the G451E mutant ($k_{cat}/K_m = 1.7 \text{ s}^{-1} \text{ mM}^{-1}$ for L-Glu production and $20 \text{ s}^{-1} \text{ mM}^{-1}$ for D-GlcN6P production) are similar to those of the wild-type enzyme (resp 3.3 and $21.25 \text{ s}^{-1} \text{ mM}^{-1}$). Thus the catalytic constants of the mutant do not explain why the gain-of-function could be linked to a higher synthesis of GlcN6P and UDP-GlcNAc. The authors agree that these small changes remain inconclusive (response to Referee 2, Discussion on page 16). As a consequence, the discussion deals with speculations about the mechanism of the G451E mutant and the role of the linker region, which are not comforted by experimental results.

We thank the reviewer for valuable feedback and for critical comments that are sure to enhance the rigor of our data and their interpretation. Particularly, we appreciate the critical comments on our structural work. However, we like to emphasize that our manuscript goes beyond structural work as it contains activity assays and *in vivo* data that provide mechanistic insights and that support our claims. In this context, the reviewer has unfortunately overlooked the UDP-GlcNAc inhibition assay we have performed (Figure 5 d). We have unequivocally shown that the G451E mutant is not inhibited by UDP-GlcNAc at doses where wild type GFAT-1 showed a clear inhibitory response. The data explaining the gain-of-function mechanism are solid and undebatable. Nevertheless, to reflect the reviewer's suggestions, we have changed the title to “Loss of GFAT-1 feedback regulation activates the hexosamine pathway that modulates protein homeostasis”

More experiments are needed to understand HP regulation at the molecular level, such as determination of the dissociation constants of UDP-GlcNAc with wild type and the G451E mutant as well as getting a crystal structure of the G451E mutant co-crystallized with UDPGlcNAc, in which the glutaminase domains are not involved in crystal contacts.

After careful reevaluation of our data, we think that a further characterization of the dissociation constants of UDP-GlcNAc with wild type or mutant GFAT-1 would not add substantially to our findings. Our kinetic data clearly show that the mutant has a greatly reduced sensitivity to UDP-GlcNAc (Figures 5 d, Supplementary Figure 5c) with an IC_{50} of approximately 5 mM compared to about 40 μM of the wild type enzyme. Furthermore, we could not detect an interaction in an ITC measurement. A previous publication by Assrir et al. indicated that standard methods are informative regarding this enzyme (“Differential titration fluorimetry, isothermal titration calorimetry and surface plasmon resonance were then also tested to analyze the interaction. However either the method turned out to be inappropriate giving unacceptable background

signal or the interaction was undetectable likely due to out-of-range affinity.“ p. 45, first paragraph discussion, reference 33). Nevertheless, we now discuss the affinity of the G451E mutant (see p. 18, top).

We agree with the reviewer that additional co-crystal structures would be nice to have. However, as part of the first round of revisions, we tried to get well-diffracting GFAT1/UDP-GlcNAc co-crystals not only from the wild type, but also from the gain-of-function mutant G451E. In poor-diffracting crystals, we saw different orientations of the glutaminase domains. However, the data quality was not sufficient for our manuscript. Nonetheless, our conclusions regarding the G451E gain-of-function mechanism are well supported by the available data.

Moreover, the structure in complex with Glu has not been correctly revisited with the view that Glu is the product and not the substrate.

Both L-Gln (substrate) and L-Glu (product) are able to bind to the glutaminase active site. Isupov et al. (reference 27) nicely describe the closed active site in the L-Glu-bound isolated glutaminase domain of bacterial GFAT (GlmS) and speculate how binding of L-Gln and L-Glu would differ. (“The ligands, glutamate and Glu-hydroxamate, bound to the protein, cause active-site closure.” p.805, first paragraph). In our structures, we cannot ignore the interactions of L-Glu with the Q-loop, which stabilizes the sealing of the active site by the Q-loop and additional changes of the R- and C-loop. In fact, L-Glu binding can induce some changes similar to L-Gln binding. As part of the revision process, we have removed the passages referring to the catalytic cycle in monomer A and have described the structural consequences of Glu-binding. These changes take place in the open conformation of monomer A, where the position of the glutaminase domain is fixed by crystal contact. Reviewer 1 agrees that this state “may represent a non-catalytically relevant conformational state.” (see comment p. 8 lines 16-18). Thus, it is contradicting to attach a great importance to the presence of the product in the active site. We treat the presence of Glu more neutrally as a ligand, which binds in the open conformation inducing some changes. Overall, classification of L-Glu as product would make the interpretation of the structural data easier, however, given these concerns, we do not think that this is possible.

I again do not support the publication of this article in Nature Communications because I think that, although the GFAT-1 structure remains interesting as the first structure of the human enzyme, this paper is not a story about elucidating the interdomain communication thought to be essential for the regulation of the enzyme and even less about understanding the mechanism of proteoprotective function of gain-of-function mutants.

We are glad about the reviewer’s interest in the first human GFAT-1 structure but disagree with the notion that we lack data on the role of the gain-of-function mechanism. The paper extends and supports the structural insights with activity assays and even *in vivo* experiments. First, we have multiple lines of evidence that support a critical role of the interdomain linker in the regulation of GFAT-1 activity. Second, the mechanism of the gain-of-function through reduced sensitivity to UDP-GlcNAc feedback inhibition is supported by solid functional data. Moreover, the *in vivo* data support the relevance of this mechanism in a mammalian system.

According to me, this paper tells a story about the conformational changes occurring during catalysis upon departure of the glutamate product (comparison of the structures of GFAT-1 in complex with Glu and Glc6P/GlcN6P or in complex with Glc6P/GlcN6P alone). Another important thing that has been shown is that the interdomain linker has

a role in UDP-GlcNAc inhibition. However, UDP-GlcNAc binding did not lead to structural changes at the glutaminase site, thus giving no explanation about the transmission of the inhibitory signal to the glutaminase active site.

In the first version of our manuscript, we expected changes at the glutaminase site upon UDP-GlcNAc binding mediated through interdomain communication. However, in the revised manuscript we refrained from this expectation and introduced the hypothesis that UDP-GlcNAc might disturb the tight coupling of the active sites by interference with the orientation of the two domains relative to each other by interactions with the interdomain linker. We have explained our reasoning in detail on page 17. In accordance with this much more plausible scenario, there is no explanation of the signal transmission. Despite the glutamate-induced conformational changes and the role of the interdomain linker in UDP-GlcNAc inhibition, we explain the gain-of-function mechanism and show the relevance of the G451E mutation and hexosamine pathway activation in mammalian cells.

Major points

Abstract: delete “owing to different interdomain linker contacts”. This does not explain why there is no inhibition of the catalytic activity. As stated above, nothing in the crystallographic results explains why UDP-GlcNAc binds to GFAT-1 whereas there is a loss of UDP-GlcNAc feedback inhibition.

Our study comprises more than crystallographic results; functional assays support our findings that the interdomain linker is important for UDP-GlcNAc inhibition (e.g. UDP-GalNAc, mutant Q307A). Prompted by the reviewer’s comment we modified the sentence to “The longevity-associated G451E variant shows drastically reduced sensitivity to UDP-GlcNAc inhibition in enzyme activity assays.”.

p. 4 line 24

after “Most insights into GFAT structure come from its bacterial homolog, glucosamine-6-phosphate synthase (GlmS)” add “whose full-length structure has been determined in complex with substrates and product”. Omitting this statement can lead to misleading understanding that only structures of the isolated domains of the bacterial enzyme have been solved (see Reviewer 2, 1st comment: Though isolated structures of domains of the enzyme from bacterial as well as eukaryotic sources are available, this is the first full length structure...)

We had already given in the first version of the manuscript full credit to previous work on the full-length bacterial enzyme, and we have added now also this statement.

P. 5 lines 4-6. “Structural and functional analyses of point mutants show that their gain-of function results from loss of UDP-GlcNAc inhibition.” This statement is wrong because the structural analysis of point mutant does not explain why the gain-of-function comes from loss of UDP-GlcNAc inhibition.

This does not hold true for the mutant G461E. The G461E mutation also causes a gain-of-function by loss of UDP-GlcNAc inhibition. Supplementary Figure 4h clearly shows in the structure how the mutated Glu side chain blocks the binding pocket and that the mutant loses the ability to bind to UDP-GlcNAc.

p. 8 lines 16-18: “This open conformation in monomer A allows conformational changes at the active sites with loop movements, which are restricted in the closed conformation in monomer B.” This is wrong to say that the open state of monomer A allows for conformational changes that cannot occur in monomer B because the conformation of monomer A (not B) is restricted by crystallographic contacts. The

conformation of monomer A thus may represent a non-catalytically relevant conformational state.

We agree that our statement was misleading and have re-phrased the sentence to “This open conformation in monomer A allows conformational changes at the active sites with loop movements, which do not occur in the closed conformation in monomer B.”.

p. 8 lines 25-28: “Compared to the *E. coli* structures, the conformation of monomer B is consistent with the first steps of the catalytic cycle, comprising initial sugar binding and structural ordering of Cloop and glutaminase domain(40)”. Or rather the B monomer, in complex with Glc6P, corresponds the last step of catalysis, just before departure of the Glc6P product, and is more like the *E. coli* GlmS/GlcN6P structure (reference 45)? And the A monomer with Glu and Glc6P bound corresponds to the preceding step, just before the Glu product has left? In that case, it is Glu departure (and not “L-Glu-induced shifts” p. 8 before last line) that triggers conformational changes of the Q and R loops. My proposal is supported by the new structure in complex with Glu and GlcN6P which is very similar to the structure in complex with Glu and Glc6P.

Prompted by the reviewer’s comment we added a section where we compare the human GlcN6P-bound structure to the *E. coli* GlcN6P-bound structure and placed it into the catalytic cycle (see p. 9, middle). However, the *E. coli* GlmS/GlcN6P structure (Mouilleron et al., reference 41 (previously reference 45)) shows a circular GlcN6P in the active site and no electron density for the glutaminase domain was observed. Thus, the human GlcN6P-bound structure is not like the *E. coli* GlmS/GlcN6P structure. Although our structures with Glc6P and GlcN6P are very similar, they cannot represent both the last step before the departure of the product (s. image below). The Glc6P-bound structure cannot be equal to the GlcN6P-bound structure, because in the absence of L-Gln the product GlcN6P cannot be formed. Moreover, it is not possible to unambiguously place the Glu-bound monomer A in the catalytic cycle (s. above third paragraph and reviewer comment p. 8 lines 16-18).

Structural catalytic scheme of GlmS (modified from Mouilleron et al. (reference 40))

p.9 lines 9-16 “These data suggest that the Q-loop covers the glutaminase active site upon L-Glu binding, but as it is the product, it cannot induce the coupling of both active sites by ammonia channel formation.” No. It means that after reaction has occurred (Glc6P/GlcN6P formation), the channel gets closed.

We disagree with the reviewer with regard to the Glu-induced changes (s. above). Moreover, in our structures no channel is formed (see p. 9 top/middle and p. 16 top); thus, no channel can get closed.

Replace “We wondered how the presence of L-Glu and GlcN6P would affect the structure and co-crystallized GFAT-1 in the presence of both product molecules.” by “The proposal that monomer A of the GFAT-1 structure in complex with Glu and Glc6P mimics GFAT-1 in complex with both products is in agreement with the structure of GFAT-1 in complex with Glu and GlcN6P...”

We do not concur with the opinion of the reviewer (s. comment “p. 8 lines 25-28”) and did not implement this change in our manuscript.

p. 9 line 18: change “Given the apparent role of L-Glu in the conformation of the Q- and R-loops, we wondered how its absence would affect the structure of GFAT-1 monomer A ” by “Given that the occupancy of the glutaminase site by Glu is linked to the conformation of the Q- and R-loops, we wanted to know the conformation of these loops in the absence of Glu”

We thank the reviewer for this comment and have implemented this suggestion in our manuscript.

p. 10 lines 4-7 I suggest to change “Together, the glutamate-free structure of monomer A suggests a communication of the glutaminase site through Arg33 with the isomerase domain.” by something like “the comparison of the structures of GFAT-1 in complex with Glu and Glc6P/GlcN6P or in complex with Glc6P/GlcN6P alone shows that the signal of glutamate departure is communicated to the isomerase domain through Arg33”.

We disagree with the reviewer with regard to the role of Glu in our structures (s. above third paragraph) and did not implement this change in our manuscript.

p. 13 line 15 delete “Understanding GFAT-1 gain-of-function by a specific point mutation“

We understand the gain-of-function mechanism in mutant G451E: the mutant loses UDP-GlcNAc feedback inhibition at doses where wild type GFAT-1 is clearly inhibited (see Figure 5d). Thus, we did not delete this statement.

p. 15 Lines 3-4 “present new mechanistic insights into its regulation that affects protein homeostasis through the HP.” I disagree.

The paper includes mechanistic insights based on a variety of structural and functional experiments. Further, we have previously shown that the specific G451E gain-of-function mutation has an effect on protein homeostasis. Our statement is true.

Legend Figure S3. Change “Glutamate determines conformation of Q-loop, R-loop, and C-loop” by “The conformations of Q-loop, R-loop, and C-loop are linked to the occupancy of the glutaminase site by Glutamate”

According to the manuscript checklist for Nature Communications figure legends should “contain a brief title”. Therefore, we prefer our short title but are open to change is according to the editor’s input.

p. 16 lines 4-8 “In contrast, no ammonia channel was formed after L-Glu binding in our structure. These data suggest that L-Glu binding induces some changes similar to L-Gln binding, but as it is the product and not substrate of the reaction, it switches Cys2 to a catalytic incompetent rotamer and does not provoke the formation of the ammonia channel.” My interpretation is that the structure with the Glu product cannot be compared with the structure with the Gln substrate. Same comment as p. 8 lines 25-28.

We disagree with the reviewer with regard to the role of Glu in our structures (s. above third paragraph). We therefore prefer to preserve this passage.

p. 16 lines 26-30 “After disruption of the salt bridge by reorientation of the R-loop upon glutamate binding, the cleft could close. This allows the C-loop to interact with both active sites, which is necessary for catalysis. Thereby, the R-loop might signal the presence of substrate in the glutaminase active site to the isomerase domain.” I would tell the story in the other way, following the catalytic cycle: which conformational changes occur upon glutamate release. I do not believe that the structure with glutamate mimics the structure with glutamine. But what seems interesting is that the authors have shown for the first time that the salt bridge between Arg33 and Asp262 breaks when glutamate leaves and a new salt bridge occurs between Arg33 and Asp667 (from the isomerase domain), perhaps preparing the release of the Glc6P/GlcN6P product?

We do not concur on the role of L-Glu (s. third paragraph) and we prefer not to place the structure in the catalytic cycle (s. above comment “p. 8 lines 25-28”). Thus, we did not implement this suggestion in our manuscript.

Discussion p. 17 and 18 “There are several lines of evidence that support a key role of the interdomain linker in this process”. The discussion is very confused because it mixes arguments that indicate that UDP-GlcNAc could function as a feedback inhibitor by altering the conformation of the linker region and thus the communication between the two active sites (which is probable) and arguments to try to explain why the feedback inhibition would be released in the G451E mutant (for which there is no mechanism that can be drawn from the present study). Also, the increased sensitivity of the Q307A mutant to UDP-GlcNAc compared to the wild-type protein is very difficult to understand at present. An important role of Q307 seems unlikely given it is not conserved (Lys in *C. elegans*, Thr in *D. melanogaster*, Ser in *A. thaliana* and *V. radiata* Figure 6B).

We thank the reviewer for pointing this out and in order to help the reader to follow our argumentation we separated the arguments that underline the role of the interdomain linker in inhibition and the hypotheses why the mutants show altered sensitivities to UDP-GlcNAc (see p. 17f). Again, we like to emphasize that we clearly show the gain-of-function mechanism for the G451E mutant, which is the loss of UDP-GlcNAc regulation. In humans, our analysis of mutant Q307A reveals a functional role of Gln307 in inhibition. Neither IC_{50} values, nor UDP-GlcNAc concentrations are known in other organisms. Thus, a variation of this position might reflect a fine tuning of the inhibitory response in different species.

p. 17 Lines 30-32 “Furthermore, in the double mutant, Glu451 might prevent the correct positioning of interdomain linker and UDP-GlcNAc-bound isomerase domain”. Not understood

We re-phrased the discussion to make this clearer: “In the double mutant, the Glu451 side chain might disturb UDP-GlcNAc binding and prevent a close interaction between the UDP-GlcNAc-bound isomerase domain and the interdomain linker.” (see p. 18, middle).

Minor points:

p. 6 line 17: delete “as” in “form as an asymmetric dimer”

We implemented this suggestion in our manuscript.

p. 7 line 11: replace “which was formed” by “which is formed”

We implemented this suggestion in our manuscript.

p. 8 lines 7-10: the sentence “The two crystallographically independent monomers form an asymmetric dimer due to the shift of the glutaminase domains relative to their respective isomerase moieties, which themselves form a symmetric assembly” is not clear. Replace by “The two crystallographically independent monomers form a dimer. The dimer is asymmetric because the two glutaminase domains are oriented differently relative to their respective isomerase moieties, which form a symmetric assembly”.

Thanks for making the language clearer. We gladly implemented this suggestion in our manuscript.

p.9 before last line replace “rotates” by “is rotated”

We implemented this suggestion in our manuscript.

p. 10 give the name of GalNAc

We implemented this suggestion in our manuscript.

p. 11 5th line replace “established” by “used” because these methods have already been published.

We implemented this suggestion in our manuscript.

It would be clearer if all the catalytic constants, including k_{cat}/K_m are given in a main Table (rather than in Fig and Supp Figures). Then, the numbers need not to be given in the main text.

We thank the reviewer for this constructive suggestion and included a table with all catalytic constants (new Table 2).

p. 11 Supplementary Figure 4h does not exist.

The Supplementary Figure 4h exists as a figure, in the figure legends and as a reference in the manuscript.

p. 11 last lines. Replace “our data support a critical functional role of the sugar’s 4-OH interaction with the interdomain linker in the feedback inhibition.” by “feedback inhibition of UDP-GlcNAc involves a critical interaction between the sugar 3OH group and the interdomain linker.”

As suggested, we reformulated the sentence to “feedback inhibition of UDP-GlcNAc involves a critical interaction between the sugar’s C4 hydroxyl group and the interdomain linker”.

Reviewer #2 (Remarks to the Author):

The manuscript entitled „Crystal structure of human GFAT-1 provides mechanistic basis for hexosamine pathways activation that modulates proteostasis” is a resubmission after revision. The authors did a tremendous amount of work to address the comments of the reviewers and improved the manuscript including additional experiments to support their claim. With a few minor tweaks, the manuscript will be more than ready for publication

We thank the reviewer for the objective judgement and constructive criticism of our work. We address the comments point-by-point below.

Title:

I feel, there should be an “a” before mechanistic.

We changed the title to “Loss of GFAT-1 feedback regulation activates the hexosamine pathway that modulates protein homeostasis”.

Introduction

Page 4:

Most insights into the GFAT structure and function.....

We implemented this suggestion in our manuscript.

Page 5:

Interfering with GFAT-1..... The authors speak about GFAT in general and specify here to GFAT-1. Why?

Prompted by the reviewer’s comment we changed GFAT-1 to GFAT. According to Oki et al. (reference 26) GFAT-1 is ubiquitously expressed, while GFAT-2 is specific for neuronal tissues. Thus, activation of GFAT-1 would result in a global effect, while GFAT-2 activation would affect more specific neuronal tissues, which would be interesting for the treatment of age-related proteinopathies.

Page 10

There are only two interactions of the glucose ... N-Acetylglucosamine

We implemented this suggestion in our manuscript.

Page 12

The authors say that in both, wt and mutant G451E UDP-GlcNAc (5 mM) could be found into the active site, though G451E is not inhibited by UDP-GlcNAc. Is that true for 5 mM in solution?

This is an excellent comment that we should have thought of earlier. To address this comment, we performed a UDP-GlcNAc inhibition assay with doses up to 5 mM. Mutant G451E showed an approximately 50% reduction in activity at 5 mM UDP-GlcNAc. Thus, the G451E mutant does not lose complete UDP-GlcNAc inhibition; however, with an IC_{50} of approximately 5 mM it responds around 125-fold less to UDP-GlcNAc than the wild type. These data do not support our conclusion that “the G451E substitution appears to uncouple the sensing of UDP-GlcNAc binding to the isomerase domain from the allosteric inhibition of the glutaminase activity by interacting with the interdomain linker.” (old discussion p. 15). Therefore, we deleted this conclusion.

Page 14

Is it really enhance activity of GFAT-1 or rather constant activity due to the lack of feedback inhibition? The kinetic data suggest the latter.

The reviewer is correct, the constant activity due to the missing UDP-GlcNAc inhibition causes the gain-of-function. Therefore, we re-phrased our conclusion (“the gain-of-function resulted from constant GFAT-1 activity due to the lack of feedback inhibition *in vivo*.”)

Page 16

Reorientation of the R-loop upon glutamine

We thank the reviewer for pointing this out and replaced the “glutamate” in the manuscript by “glutamine”.

Page 17 and 18

The authors cite Assrir et al. that UDP or UDP-Glc does not inhibit GFAT-1. Taken into account that the changes upon binding or not near the sugar, it makes sense to observe the changes for GlcNAc and GalNAc, emphasizes that the UDP moiety is only for binding and maybe orientation of the HexNac unit.

Is there anything visible from the structures (changes in the loops), interactions etc., which could hint to a possible cause of the interactions with the sugar part? Q307 was suggested to interact with the HexNAC unit, as far as I understood.

From the current structures we did not observe any interactions, loop changes etc., which would help to understand the specificity of GFAT-1 to the GlcNAc-moiety. In order to get further structural insights, we solved the structure of mutant Q307A with and without UDP-GlcNAc. However, the structures resemble the wild type. Since the glutaminase domain and interdomain linker are very flexible relative to the isomerase domain, missing evidence from the structural data do not exclude interactions *in vitro*.

From previous pages I got the impression that the correct orientation and tight coupling is necessary for efficient catalysis for example the formation of the ammonia channel. But on page 17 the authors say that Q307 and E451 prevent close interactions. That sounds contradictory. I would expect that both residues stabilize a closer conformation and Q307A might even allow for a closer and tighter conformation?

We agree that our wording was misleading and thank the reviewer for pointing out this weakness. We have to differentiate between the correct orientation of the two domains important for catalysis and the orientation of the UDP-GlcNAc-bound isomerase and the interdomain linker relevant for UDP-GlcNAc inhibition. For catalysis, both active sites have to be coupled, thus the two domains interact at the interface of their active sites with each other. In UDP-GlcNAc inhibition, the UDP-GlcNAc-bound isomerase interacts with the interdomain linker and not at the active sites. We have re-phrased the discussion to clarify these two things (see p. 18, top).

Figures.

Fig2 a-e) (according to legend) shows only panels a-d

Thanks for pointing out this mistake, it is corrected in the figure legends.

Fig. 4 What is the role of the shown Magnesium? It seems too far away to be important for the interaction with the nucleotide?

We can exclude a functional role of the Magnesium because the inhibition assays were performed in the presence of 1 mM EDTA. Thus, the presence of the metal ion in the UDP-GlcNAc-bound structures is rather an observation than a functionally important finding.

Fig 4c) This is not essential, but to engage in two H bonds Q310 should have a switched side chain. Might be supported by B-factors?

We thank the reviewer for pointing this out; unfortunately, B-factors do not support the orientation of the side chain. However, MolProbity suggests the current orientation and we decided to stay with this conformation.

Supplementary Material:

Supp Fig1 exists twice (numbering issue)

We will make sure with the editor that the figures are numbered properly.

REVIEWERS' COMMENTS:

Reviewer #2 (Remarks to the Author):

The manuscript "Loss of GFAT-1 feedback regulation activates the hexoasamine pathway that modulates protein homeostasis" by Ruegenberg et al. is a revision to address the concerns raised by reviewers in the previous round.

I believe the manuscript is ready for publication, even if it does not answer all questions, which might arise in the context of the observed data. However the structure and biochemical data are solid and shows possible reasons for the observed lack of feedback inhibition of one point mutation, though a definite answer cannot be given with the current available data.

Some minor changes may be incorporated by the authors:

P4 Most insights....comes from ist.....

P7 while beta sheets.....are more extended in the human enzyme. The sentence has a repeat in regard of the beta sheets and it would be better to say beta strands. That is just semantic beta strands form the sheet. The strands are longer and one gets a larger sheet structure.

Phosphorylated in both,.....

P9 Further, the hydrogen bonds ◊ Furthermore,.....

REVIEWERS' COMMENTS:

Reviewer #2 (Remarks to the Author):

The manuscript "Loss of GFAT-1 feedback regulation activates the hexoasamine pathway that modulates protein homeostasis" by Ruegenberg et al. is a revision to address the concerns raised by reviewers in the previous round.

I believe the manuscript is ready for publication, even if it does not answer all questions, which might arise in the context of the observed data. However the structure and biochemical data are solid and shows possible reasons for the observed lack of feedback inhibition of one point mutation, though a definite answer cannot be given with the current available data.

We thank the reviewer for the valuable comments throughout the revision process that have contributed to the quality of the data we present in the paper.

Some minor changes may be incorporated by the authors:

P4 Most insights....comes from ist.....

We think that the grammar in our sentence is correct and have not made a change here.

P7

while beta sheets.....are more extended in the human enzyme. The sentence has a repeat in regard of the beta sheets and it would be better to say beta strands. That is just semantic beta strands form the sheet. The strands are longer and one gets a larger sheet structure.

We have made the suggested change.

Phosphorylated in both,.....

We have made the suggested change.

P9 Further, the hydrogen bonds Furthermore,.....

We have made the suggested change.